# RockTS: Robust Time Series Forecasting based on Information Bottleneck and Optimal Transport

## Abstract

Time series forecasting plays a crucial role in numerous real-world applications. Existing works mostly assume clean and regular historical sequences for predicting future ones. However, real-world time series data often contain anomalous subsequences that deviate from the regular patterns of the entire series, posing challenges to accurate forecasting. In this paper, we propose RockTS, a novel end-to-end framework for robust time series forecasting based on Information Bottleneck and Optimal Transport, which integrates the detection and imputation of anomalous subsequences into the forecasting task through a unified optimization objective. RockTS first introduces a detection process for anomalous patterns based on Information Bottleneck, which compresses representations of time series while retaining the information more relevant for effective forecasting. It then imputes the detected anomalous regions with normal patterns through a novel reconstruction strategy based on Optimal Transport for forecasting. Experiments on multiple real-world and synthetic datasets demonstrate that RockTS achieves superior robustness and forecasting performance.

## 1 Introduction

Time series forecasting plays a crucial role in extensive real-world applications, such as weather forecasting, energy management, financial investment, and traffic flow estimation. Deep learning models have achieved remarkable success in time series forecasting tasks. Models based on multilayer perceptrons (MLP) (Xu et al., 2024; Zeng et al., 2023), convolutional neural networks (CNN) (Wu et al., 2022; Luo & Wang, 2024), and Transformers Chen et al. (2024); Liu et al. (2024) have been continuously emerging. They achieve predictions of the future by learning complex patterns and dependencies in historical time series data.

Real-world time series data may contain anomalous subsequences that deviate from the normal patterns of the entire series occasionally Schmidl et al. (2022), due to sensor failures, transmission disturbances, malicious attacks, etc. Indeed, anomalous subsequences may lead to challenges in time series forecasting. As shown in Figure 1(a), anomalous subsequences in the historical data make the model misjudge the patterns of the time series, which in turn leads to significant prediction errors. Unfortunately, most existing robust time series forecasting methods Wang et al. (2023b); Fraikin et al. (2024); Wang et al. (2022) primarily focus on addressing issues of point-wise anomalies or distribution shift. They address point-wise anomalies by using robust loss functions and sample selection strategies for specific kinds of anomalies Cheng et al. (2024). Alternatively, they address distribution shifts by integrating a self-adaptation stage prior to forecasting Arik et al. (2022). However, anomalous subsequences are more complex compared with point-wise anomalies Cheng et al. (2024), as they display diverse lengths or patterns. Further, the anomalous subsequences may fall within the same probability distribution Nam et al. (2024); Paparrizos et al. (2022) as the entire series. Therefore, these methods struggle to withstand the interference caused by anomalous subsequences. Consequently, addressing the forecasting challenge in the context of data with anomalous subsequences becomes highly significant.

To address the negative impact of such complex and diverse anomalous subsequences on time series forecasting, an intuitive strategy is to perform an additional data cleaning process to filter the

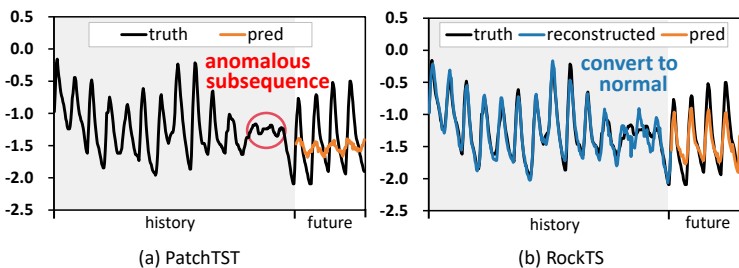

Figure 1: Example of the prediction results on ETTh2 of PatchTST. RockTS converts the anomalous subsequence to normal and uses the same predictor as PatchTST.

anomalous subsequences in the data before they enter the forecasting model Li et al. (2022); Bohlke-Schneider et al. (2020). However, such strategy models the data cleaning task and the prediction task separately and independently, failing to fully consider the specific requirements of the forecasting task when cleaning data. As a result, they struggle to accurately capture anomalous subsequences that are detrimental to the forecasting task, and their cleaning process may potentially hinder the forecasting process as they may introduce additional noise for prediction Cheng et al. (2024).

To achieve robust time series forecasting on data that may contain anomalous subsequences, we propose a novel end-to-end framework based on Information Bottleneck and Optimal Transport (RockTS), for robust time series forecasting. The main idea of RockTS is to integrate the detection and imputation for anomalous subsequences into forecasting tasks using a unified optimization objective. It first detects the anomalous subsequences in time series using a novel adaptive detector based on Information Bottleneck (IB) Tishby & Zaslavsky (2015), and then imputes the detected regions into the normal patterns by a reconstruction strategy based on Optimal Transport (OT). Finally, the imputed series is fed into the predictor for the collaborative learning of detection, imputation, and forecasting tasks.

To integrate anomalous subsequence detection in the forecasting process, RockTS innovatively introduces an adaptive detector based on IB to locate the anomalous subsequences in time series through masking from the data. To address the issue of insufficient consideration of prediction requirements in the normal detection process, we leverage IB to compress representations while retaining relevant information for effective prediction. We optimize the detection process by: 1) minimizing mutual information between the original and the remaining series after masking to filter regions useless for prediction, while 2) maximizing mutual information between the remaining series after masking and future series to retain forecasting-relevant regions. This trade-off optimizes the locations to cover anomalous subsequences that negatively affect the prediction, thus allowing the detector to efficiently locate the anomalous subsequences by the mask locations.

To better impute anomalous subsequence regions in the forecasting process, preserving the patterns of the original time series and avoiding the re-emergence of anomalous subsequences are two critical points. We innovatively propose a reconstruction strategy based on OT to achieve this. We first use a reconstruction network to impute the detected regions with continuous values. Then, we use a transport matrix that models the correlation among time points to further adjust these imputed time series, such that they preserve the patterns of the original time series. Moreover, we set a higher transport cost for the detected regions to constrain the transport matrix, to prevent the re-emergence of anomalous subsequences caused by over-optimization of the reconstruction loss.

Our contributions can be summarized as follows:

- We propose RockTS, a novel end-to-end framework for robust time series forecasting that for the first time directly addresses the issue of anomalous subsequences in time series.

- We introduce an adaptive detector based on the information bottleneck to detect the anomalous subsequences in the forecasting process, and retain forecasting-relevant regions.

- We design a reconstruction strategy based on OT to impute the masked regions into the normal patterns in forecasting, while avoiding the re-emergence of anomalous subsequences.

- We apply RockTS on multiple real-world datasets and synthetic datasets injected with anomalous subsequences. RockTS exhibits strong robustness and superior forecasting performance, withstanding interference from various types of anomalous subsequences.

## 2 RELATED WORK

### 2.1 TIME SERIES FORECASTING

Time series forecasting primarily involves predicting future sequences based on historical sequences. Statistical methods, such as ARIMA Box & Jenkins (1968) and VARSims (1980), mainly capture simple temporal patterns but struggle with modeling complex dependencies, which limits their prediction performance. In contrast, deep learning methods have rapidly advanced due to the neural networks' powerful ability to model complex patterns. RNN-based and CNN-based methods focus on capturing local temporal dependencies Wu et al. (2022); Wang et al. (2023a); Flunkert et al. (2017); Lin et al. (2023). Transformer-based methods, leveraging the global modeling capability of Attention mechanisms, excel at capturing complex and long-term temporal dependencies. Methods like Informer Zhou et al. (2021), Autoformer Wu et al. (2021), and Triformer Cirstea et al. (2022) reduce the time and space complexity of Attention from quadratic to linear. Others focus on modeling temporal characteristics such as non-stationarity Liu et al. (2022), frequency Zhou et al. (2022), multi-scale patterns Chen et al. (2024), and channel correlations Liu et al. (2024). MLP-based methods have gained attention due to their lightweight architectures and high efficiency Zeng et al. (2023); Xu et al. (2024); Wang et al. (2024). However, existing forecasting methods are significantly impacted by anomalous subsequences, often leading to a misjudgment of time series patterns. RockTS adaptively detects anomalous subsequences and transforms them into normal patterns that are conducive to accurate forecasting.

### 2.2 ROBUST TIME SERIES FORECASTING

Real-world time series often exhibit various types of anomalous patterns, making robust time series forecasting a mainstream approach to address it Cheng et al. (2024); Arik et al. (2022); Fraikin et al. (2024); Kim et al. (2025). For robust forecasting against point anomalies, RobustTSF Cheng et al. (2024) identifies informative samples by evaluating the variance between the original input time series and its trend component, followed by employing a robust loss function to improve the forecasting process. For distribution shift robustness, several innovative mechanisms have been proposed to mitigate distribution shifts caused by the non-stationarity of time series, offering better adaptability to changing data distributionsArik et al. (2022); Wang et al. (2022). For robust forecasting with missing data, T-Rep Fraikin et al. (2024) enhances model resilience by learning temporal embeddings and leveraging pretraining techniques to address the challenges posed by missing values. However, existing robust time series forecasting methods primarily target simple scenarios, such as point anomalies or basic distribution shifts, and struggle to handle complex and diverse anomalous subsequences. To address this limitation, RockTS, based on Information Bottleneck and Optimal Transport (OT) theory, adaptively detects and imputes anomalous subsequences during the forecasting process, significantly enhancing the robustness of predictions.

## 3 PRELIMINARIES

**Problem Formulation.** Given the historical time series $\mathbf{x} = \{x_1, ..., x_L\}$, with $x_i \in \mathbb{R}$ denoting the observation at the timestamp $i$, and $L$ is the size of look-back window, the goal of time series forecasting is to predict the future values $\mathbf{y} = \{x_{L+1}, ..., x_{L+F}\}$, where $F$ is the forecast horizon.

In this paper, we investigate algorithms for robust time series forecasting. We define an anomalous subsequence as $\mathbf{x}_{s,e} = \{x_s, ..., x_e\}$ with length $e - s + 1 \geq 1$ that deviates from the normal patterns in $\mathbf{x}$. The objective of robust time series forecasting is to accurately predict $\mathbf{y}$ even when the input data $\mathbf{x}$ contains possible anomalous subsequences $\{\mathbf{x}_{s_i,e_i}\}_{i=1}^n$, where $n$ is the number of anomalous subsequences.

## 4 METHODOLOGY

To achieve robust time series forecasting on data that may contain anomalous subsequences, we propose a robust time series forecasting model based on Information Bottleneck and Optimal Transport, RockTS. As shown in Figure 2, RockTS employs a detect-impute-forecast workflow. Initially, an adaptive detector based on IB is used to identify the anomalous subsequences within the time series, and then the time series is masked based on the detection results. Next, the masked sequence is imputed by a reconstruction module based on OT, generating the reconstructed series. Finally, the reconstructed series is fed into the prediction module to generate the prediction results.

Throughout this workflow, RockTS integrates the detection, imputation, and prediction tasks into a unified optimization objective. In the following parts, we describe robust forecasting based on IB and reconstruction based on OT in detail.

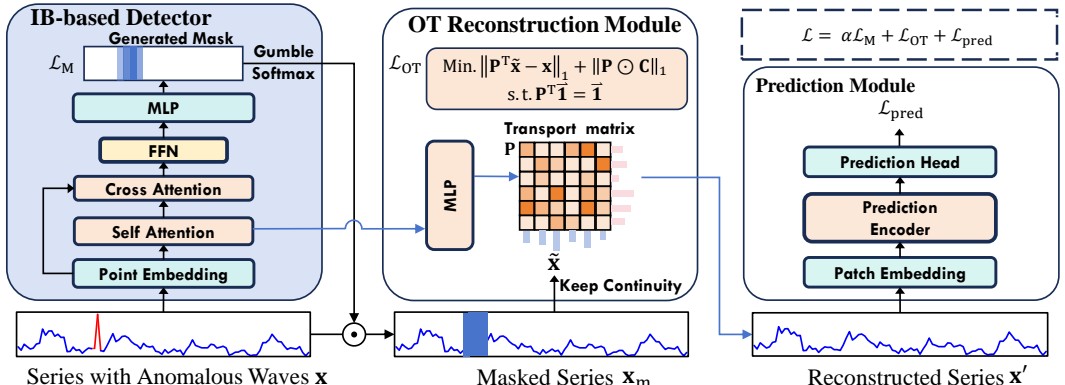

Figure 2: Overall structure of RockTS. RockTS employs a detect-impute-forecast workflow that uses the IB-based detector to identify the anomalous subsequences in time series, and then uses the OT reconstruction module to impute these detected regions. Finally, the prediction module is used to generate the prediction series. The detection, imputation, and prediction tasks are integrated into a unified optimization objective by this framework.

### 4.1 ROBUST FORECASTING BASED ON INFORMATION BOTTLENECK

**Information Bottleneck Based Detector.** Anomalous subsequences in time series display diverse lengths and patterns, and may be within the same distribution as normal series, which poses a significant challenge for direct detection. They can affect the model's analysis of the time series' complex patterns and dependencies, which limits the prediction performance.

To eliminate the effect of anomalous subsequences on the prediction, we propose an IB-based detector to identify these anomalous subsequences by learning to mask them from data. To achieve this, we employ a perturbation mask $\mathbf{M} = \{m_i\}_{i=1}^{L}$ to extract a masked series $\mathbf{x}_{\mathrm{m}}$, where $m_i \in \{0, 1\}$ and $L$ denotes the length of the input data $\mathbf{x}$, and the masked series $\mathbf{x}_{\mathrm{m}} = \mathbf{x} \odot \mathbf{M}$, where $\odot$ means the elements multiplication. With the concept that an optimal representation contains minimal original information from the input but keeps sufficient relevant information necessary for the forecasting task, we try to find a compressed $\mathbf{x}_{\mathrm{m}}$ by masking irrelevant or anomalous subsequences from the original series $\mathbf{x}$, encapsulating only the information useful for the future series $\mathbf{y}$. Formally, the objective of finding the optimal masked series $\mathbf{x}_{\mathrm{m}}$ is defined by:

$$\mathbf{x}_{\mathrm{m}}^{*} := \underset{\mathbb{P}(\mathbf{x}_{\mathrm{m}}|\mathbf{x})}{\arg\min} \, \alpha \, \underbrace{I\left(\mathbf{x}; \mathbf{x}_{\mathrm{m}}\right)}_{\text{Compactness}} - \underbrace{I\left(\mathbf{y}; \mathbf{x}_{\mathrm{m}}\right)}_{\text{Predictiveness}}, \tag{1}$$

where $I(.\,;.)$ denotes mutual information between series pairs and $\alpha$ is a hyperparameter that governs the trade-off between minimality and sufficiency constraints. The first part in Equation 1 is the compactness term to optimize the masking process, ensuring the thorough filtering of anomalous subsequences useless for forecasting. The latter part is the predictiveness term, which constrains the retention of forecasting-relevant information. Note that $I(\mathbf{y}; \mathbf{x}_{\mathrm{m}}) = H(\mathbf{y}) - H(\mathbf{y}|\mathbf{x}_{\mathrm{m}})$, where the entropy $H(\mathbf{y})$ is a statistic of time series and remains constant. Therefore, the process of maximizing the mutual information $I(\mathbf{y}; \mathbf{x}_{\mathrm{m}})$ between the compressed masked series $\mathbf{x}_{\mathrm{m}}$ and the target future series $\mathbf{y}$ can be reformulated as minimizing the conditional entropy $H(\mathbf{y}|\mathbf{x}_{\mathrm{m}})$ of $\mathbf{y}$ given $\mathbf{x}_{\mathrm{m}}$:

$$\mathbf{x}_{\mathrm{m}}^{*} := \underset{\mathbb{P}(\mathbf{x}_{\mathrm{m}}|\mathbf{x})}{\arg\min} \, \alpha \, I\left(\mathbf{x}; \mathbf{x}_{\mathrm{m}}\right) + H\left(\mathbf{y}|\mathbf{x}_{\mathrm{m}}\right). \tag{2}$$

**Learning to Mask by Compactness and Predictiveness.** The compactness term $I\left(\mathbf{x}; \mathbf{x}_{\mathrm{m}}\right)$ aims to ensure that the masked series contains the necessary information for the prediction with minimal information retained, thereby removing anomalous subsequences that are irrelevant to the prediction or even negatively impact prediction performance. However, directly optimizing mutual information $I\left(\mathbf{x}; \mathbf{x}_{\mathrm{m}}\right)$ may lead to the retaining of a significant amount of low-entropy components in $\mathbf{x}_{\mathrm{m}}$ that

contains less information, which is inconsistent with our desired objective of retaining more high-entropy components that contains necessary information. Therefore, directly optimizing mutual information cannot achieve the goal of compactness.

To address this problem, we consider the upper bound to simplify the optimization objective based on it.

$$I\left(\mathbf{x};\mathbf{x}_{\mathrm{m}}\right) \leq \mathbb{E}_{\mathbf{x}}[D_{\mathrm{KL}}[\mathbb{P}_{\theta}(\mathbf{x}_{\mathrm{m}}|\mathbf{x})||\mathbb{Q}(\mathbf{x}_{\mathrm{m}})]], \tag{3}$$

where $D_{\mathrm{KL}}$ is the Kullback–Leibler divergence. The derivation for the upper bound is detailed in Appendix B. Additionally, we define $p_{\theta} \sim \mathbb{P}_{\theta}$ as the detector with parameter $\theta$ which generates a vector of probabilities to extract the proper $\mathbf{x}_{\mathrm{m}}$ from $\mathbf{x}$. Further, the $\mathbb{Q}(\mathbf{x}_{\mathrm{m}})$ is defined as the prior distribution to regulate the detector. With these definitions established, the upper bound can be further simplified as discussed in the following paragraph.

Specifically, we train an extractor $p_{\theta}$ to generate a vector of probabilities $\lambda = p_{\theta}(\mathbf{x}) \in [0,1]^L$, where each element $\lambda_i$ corresponds to the probability of retaining the corresponding element of $\mathbf{x}$ in $\mathbf{x}_{\mathrm{m}}$ and $m_i \sim \mathrm{Bernoulli}(\lambda_i)$. Further, we define $\mathbb{Q}\{\mathbf{M}\}$ as the Bernoulli distribution with a sparsity parameter $\tau \in (0,1)$, which regulates the generation of $\lambda$, aligning it with the prior distribution $\mathbb{Q}\{\mathbf{M}\} \sim \prod_{i=1}^{L} \mathrm{Bernoulli}(\tau)$. Thus, we transform the problem of obtaining $\mathbf{x}_{\mathrm{m}}$ into generating forecasting-relevant attribution scores $\lambda$ by optimizing $\theta$. The original compactness term term in Equation 1 is transformed into a more tractable loss $\mathcal{L}_{\mathrm{M}}$ as follows:

$$\mathcal{L}_{\mathrm{M}} = \mathbb{E}_{\mathbf{x}}[D_{\mathrm{KL}}[\mathbb{P}_{\theta}(\mathbf{M}|\mathbf{x})||\mathbb{Q}(\mathbf{M})]] = \sum_{i=1}^{L}\left[\lambda_i \log\left(\frac{\lambda_i}{\tau}\right) + (1-\lambda_i)\log\left(\frac{1-\lambda_i}{1-\tau}\right)\right]. \tag{4}$$

The loss $\mathcal{L}_{\mathrm{M}}$ effectively limits the average number of non-zero elements in the mask while avoiding the destruction of compactness caused by directly minimizing mutual information. To prevent the mask $\mathbf{M}$ that causes $\mathbf{x}_{\mathrm{m}}$ to be discontinuous, we further introduce a continuity term to enhance the continuity of $\mathbf{x}_{\mathrm{m}}$, thus the final loss for compactness is:

$$\mathcal{L}_{\mathrm{M}} = \sum_{i=1}^{L}\left[\lambda_i \log\left(\frac{\lambda_i}{\tau}\right) + (1-\lambda_i)\log\left(\frac{1-\lambda_i}{1-\tau}\right)\right] + \frac{1}{L}\cdot\sum_{i=1}^{L-1}\sqrt{(\lambda_{i+1}-\lambda_i)^2}, \tag{5}$$

where the second part is the continuity term.

As described in Equation 2, the predictiveness term is defined as minimizing the conditional entropy $H\left(\mathbf{y}|\mathbf{x}_{\mathrm{m}}\right)$, which is equivalent to maximizing the conditional probability $P(\mathbf{y}|\mathbf{x}_{\mathrm{m}})$. Depending on different assumptions regarding the error distribution, this objective can be transformed into minimizing either Mean Squared Error (MSE) or Mean Absolute Error (MAE) between the prediction $\hat{\mathbf{y}}$ obtained from $\mathbf{x}_{\mathrm{m}}$ and ground truth $\mathbf{y}$ (Bishop & Nasrabadi, 2006). In alignment with current mainstream practices in the field, we adopt MSE as the optimization loss for the predictiveness term in Equation 1.

$$\mathcal{L}_{\mathrm{pred}} = ||\mathbf{y} - \hat{\mathbf{y}}||_F^2 \tag{6}$$

**Implementation of Framework.** We establish a framework to learn the detector $p_{\theta}$ that encodes the input $\mathbf{x}$ into a score vector $\lambda$, to parameterize the stochastic mask $\mathbf{M}$. The specific implementation process is as follows: we first map the input sequence $\mathbf{x}$ to $\mathbf{Z} \in \mathbb{R}^{L \times D}$ through a linear transformation, where $D$ denotes the hidden dimensions. We then feed $\mathbf{Z}$ into self-attention to capture the long-term dependencies. Leaving out attention head indices for brevity, let $\mathbf{Q} = \mathbf{W}_{\mathrm{Q}}\mathbf{Z}$, $\mathbf{K} = \mathbf{W}_{\mathrm{K}}\mathbf{Z}$ and $\mathbf{V} = \mathbf{W}_{\mathrm{V}}\mathbf{Z}$ be the transformed query, key and value matrices, where $\mathbf{W}_{\mathrm{Q}}, \mathbf{W}_{\mathrm{K}}$ and $\mathbf{W}_{\mathrm{V}} \in \mathbb{R}^{D \times D}$. Thus the attention matrix $\mathbf{A} \in \mathbb{R}^{L \times L}$ that describing the relationships among time-point features and hidden features $\mathbf{E} \in \mathbb{R}^{L \times D}$ are given by:

$$\mathbf{A} = \mathrm{Softmax}(\frac{\mathbf{Q}\mathbf{K}^{\mathrm{T}}}{\sqrt{D}}), \mathbf{E} = \mathbf{A}\mathbf{V}. \tag{7}$$

Masks for anomalous patterns are generated by calculating the similarity between $\mathbf{E}$ and $\mathbf{Z}$ with cross attention. The hidden features $\mathbf{E}$ are used as the query, and the input $\mathbf{Z}$ is used as the key and value for cross-attention. Then, a linear transformation followed by a sigmoid function is applied to convert it into a score matrix $\lambda \in \mathbb{R}^L$:

$$\lambda = \sigma(\mathrm{CrossAttention}(\mathbf{E}, \mathbf{Z})\mathbf{W}_{\mathrm{b}}^{\mathrm{T}}), \tag{8}$$

where $\mathbf{W}_b \in \mathbb{R}^{1 \times D}$ and $\sigma()$ is the sigmoid function. Further, to generate a deterministic binary mask from the stochastic probabilities $\lambda$ and enable end-to-end optimization, we apply the Gumbel-Softmax with Straight-Through Estimation and get the hard mask $\mathbf{M}$:

$$\mathbf{M} = \text{GumbelSoftmax}(\lambda). \tag{9}$$

Then we obtain the masked sequence $\mathbf{x}_m = \mathbf{x} \odot \mathbf{M}$. Next, with the reconstruction strategy based on OT, we impute $\mathbf{x}_m$ to keep its continuity, which is described in detail in Section 4.2. Thus, we transform the $\mathbf{x}_m$ into reconstructed series $\mathbf{x}'$ with normal patterns.

To derive the prediction of future series, the reconstructed series $\mathbf{x}'$ is segmented into $N_p$ non-overlapping patches $\mathbf{x}' \in \mathbb{R}^{N_p \times L_p}$, where $L_p = L/N_p$ is length of patches. The patches $\mathbf{x}'$ are mapped to the latent space dimension $D$ through a linear projection $\mathbf{W}_p$ and add a learnable position embedding $\mathbf{W}_{pos}$ to generate $\mathbf{Z}' \in \mathbb{R}^{N_p \times D}$:

$$\mathbf{Z}' = \mathbf{W}_p \mathbf{x}' + \mathbf{W}_{pos} \tag{10}$$

Then, $\mathbf{Z}'$ will be fed into a vanilla Transformer encoder, which includes a multi-head attention block, BatchNorm layers, and a feed-forward network with residual connections. Based on the representation from the encoder, we finally use a flatten layer with a linear head to obtain the prediction result $\hat{\mathbf{y}} \in \mathbb{R}^F$, which is used to compute the prediction loss in Equation 6 during training.

## 4.2 COST-AWARE RECONSTRUCTION STRATEGY BASED ON OT

In the Section 4.1, we propose an adaptive detector to detect the anomalous subsequences in time series and generate a masked series $\mathbf{x}_m$. However, the masked series is incomplete, and the empty values disrupt the continuity of the time series, affecting the model's ability to learn the patterns of time series and leading to unstable results. Therefore, it is necessary to impute it before prediction. RockTS uses reconstruction to impute the masked time series. To impute the masked regions into normal patterns while preventing the re-emergence of anomalous subsequences, we build an OT problem for the reconstruction strategy and optimize this OT problem via a neural network. Thus the model can learn the reconstruction strategy for the masked sequences end-to-end.

**OT-Based Reconstruction.** We formulate the optimization process for imputation as an optimal transport problem. First, to restore the continuity of the masked series $\mathbf{x}_m$, We transform $\mathbf{x}_m$ through a network $\mathcal{G}$ combining a Transformer encoder and a linear head, and denote the result as $\widetilde{\mathbf{x}}$. We take the distribution of reconstruction series $\widetilde{\mathbf{x}} \in \mathbb{R}^L$ as the source distribution and take the distribution of original series $\mathbf{x}$ as the target distribution. The OT problem sets a transport strategy $\mathbf{P} \in \mathbb{R}^{L \times L}$ to transform the source distribution to the target distribution to make the reconstructed series has the same overall information as the original series with a minimum cost $||\mathbf{P} \odot \mathbf{C}||_1$, where $\mathbf{C} \in \mathbb{R}^{L \times L}$. $\mathbf{P}_{i,j}$ denotes the ratio of $\widetilde{\mathbf{x}}_i$ transporting to $\mathbf{x}_j$, and $\mathbf{C}_{i,j}$ denotes the cost of transporting from $\widetilde{\mathbf{x}}_i$ to $\mathbf{x}_j$. Thus $\mathbf{P}^T \widetilde{\mathbf{x}}$ denotes the distribution after applying the transport strategy $\mathbf{P}$ to $\widetilde{\mathbf{x}}$, which should be close to $\mathbf{x}$, and the sum of each row of $\mathbf{P}$ should be 1. Thus, we formulate $||\mathbf{P}^T \widetilde{\mathbf{x}} - \mathbf{x}||_1$ as an optimization goal and the $\mathbf{P}^T \vec{1} = \vec{1}$ as a constraint in this OT problem. To reconstruct series that do not contain anomalous subsequences, we set $\mathbf{C}$ as follows:

$$\mathbf{C}_{i,j} = \begin{cases} 1 - \lambda_j, & \mathbf{M}_j = 0 \\ 0, & \mathbf{M}_j = 1 \end{cases} \tag{11}$$

where $\mathbf{M}$ is the mask vector generated from the IB-based detector. The cost matrix $\mathbf{C}$ assigns costs exclusively to regions detected as anomalous subsequences. Specifically, the cost assigned to transporting to a region increases with the probability that the detector identifies it as an anomalous subsequence. This approach effectively suppresses the recovery of original patterns in regions identified as anomalous during the reconstruction process. Thus we formulate an OT problem as:

$$\min \ \beta \, ||\mathbf{P} \odot \mathbf{C}||_1 + ||\mathbf{P}^T \widetilde{\mathbf{x}} - \mathbf{x}||_1,$$
$$\text{s.t. } \mathbf{P}^T \cdot \vec{1} = \vec{1}, \tag{12}$$

where $\beta$ is a hyperparameter belonging to $[0, 1]$, and $\vec{1}$ denotes a unit vector with length $L$.

**Optimization of OT-based Reconstruction.** We propose learning to optimize the above OT problem by constructing a neural network to get a suitable transport strategy $\mathbf{P}$. We use $\widetilde{\mathbf{x}}$ obtained by $\mathbf{x}_m$ through network $\mathcal{G}$ as the source distribution for transport. To get a better transport strategy $\mathbf{P}$

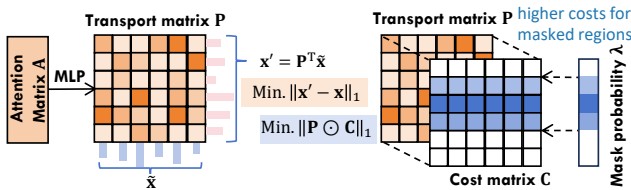

Figure 3: The transport matrix $\mathbf{p}$ is obtained by transforming attention matrix $\mathbf{A}$. The optimization objectives for OT-based reconstruction is minimizing the cost of transport and the distance between the transported distribution and source distribution.

that can make the transport result conform to the normal pattern of the time series, we model the correlation between the time points, as shown in Figure 3. In Section 4.1, we have obtained the attention matrix $\mathbf{A}$ that models the relationships and importance between time-point features. We non-linearly transform it by an MLP-network $\mathcal{H}$ to generate the transport matrix $\mathbf{P}$.

We use $\mathbf{P}$ to migrate the elements in $\widetilde{\mathbf{x}}$, and get the reconstructed series $\mathbf{x}' \in \mathbb{R}^L$. We take $\mathbf{x}'$ as an input to the predictor to get forecasting results as discussed in Section 4.1. To meet the constraint in Equation 12, we use softmax for $\mathbf{P}$, and the transformation process is specified as:

$$\mathbf{x}' = \text{softmax}(\mathbf{P})^{\mathrm{T}} \widetilde{\mathbf{x}}. \tag{13}$$

We use the MAE loss to optimize the transport result $\mathbf{x}'$ close to the target distribution $\mathbf{x}$. To avoid the anomalous subsequences in $\mathbf{x}$ being reproduced in $\mathbf{x}'$ due to over-optimization of the reconstruction loss, we calculate the total cost of the transport process and include it in the loss function. Therefore, we can introduce the optimization objective of the OT problem to the loss function:

$$\mathcal{L}_{\mathrm{OT}} = \frac{1}{L} \sum_{i=0}^{L} |\mathbf{x}_i - \mathbf{x}'| + \beta \sum_{i=0}^{L} \sum_{j=0}^{L} \mathbf{P}_{i,j} \odot \mathbf{C}_{i,j}. \tag{14}$$

### 4.3 OVERALL LEARNING OBJECTIVE.

RockTS is optimized end-to-end and the learning objective is trained by minimizing the total loss:

$$\mathcal{L} = \alpha \mathcal{L}_{\mathrm{M}} + \mathcal{L}_{\mathrm{OT}} + \mathcal{L}_{\mathrm{pred}} \tag{15}$$

where $\alpha \in [0, 1]$ is a hyperparameter for multi-loss balance. In summary, the philosophy of RockTS is that when a subsequence is anomalous that is useless for prediction, we mask it and impute it into normal pattern. When a subsequence is in normal mode, RockTS keeps its information for prediction tasks. The detection and imputation processes are trained end-to-end together with the prediction task, enabling the learning of forecasting-relevant information in the framework.

## 5 EXPERIMENTS

### 5.1 EXPERIMENTAL SETUP

**Evaluation Datasets.** To conduct comprehensive and fair comparisons for different models, we conducted experiments on eight well-known forecasting benchmarks as the evaluation datasets, including Weather, Traffic, Electricity, Solar, and ETT (4 subsets). In addition, to further evaluate the robustness of the model on data that contains anomalous subsequences, we construct synthetic datasets by injecting six types of anomalous subsequences: vmirror, hmirror, scale, outlier, noise and pattern into above real-world datasets. The specific descriptions of these anomalous subsequences are detailed in the Appendix A.2.

**Baselines.** We compare our model with nine state-of-the-art models for comprehensive evaluations, including Transformer-based models: iTransformer Liu et al. (2024), PatchTST Chen et al. (2024) and PathFormer Nie et al. (2023); CNN-based model: TimesNet Wu et al. (2022) and ModernTCN Luo & Wang (2024); MLP-based models: FITS Xu et al. (2024), TIDE Das et al. (2023), TimerMixer Wang et al. (2024) and DLinear Zeng et al. (2023).

**Implementation Details.** Consistent with previous works Nie et al. (2023), we adopted Mean Squared Error (MSE) and Mean Absolute Error (MAE) as evaluation metrics. We use the lookback window length $L = 512$ and predict the future values with lengths $F = \{96, 192, 336, 720\}$.

Table 1: The average results of four prediction lengths in real-world datasets.

| Models | ours | | ITransformer | | PatchTST | | Pathformer | | TimesNet | | ModernTCN | | Dlinear | | TiDE | | FITS | | TimeMixer | |
|---|---|---|---|---|---|---|---|---|---|---|---|---|---|---|---|---|---|---|---|---|
| Metric | MSE | MAE | MSE | MAE | MSE | MAE | MSE | MAE | MSE | MAE | MSE | MAE | MSE | MAE | MSE | MAE | MSE | MAE | MSE | MAE |
| ETTm1 | **0.345** | **0.368** | 0.362 | 0.391 | _0.349_ | 0.381 | 0.357 | _0.375_ | 0.490 | 0.464 | 0.361 | 0.430 | 0.357 | 0.379 | 0.360 | 0.381 | 0.357 | 0.377 | 0.356 | 0.380 |
| ETTm2 | **0.248** | **0.303** | 0.269 | 0.329 | 0.256 | 0.314 | _0.253_ | _0.309_ | 0.317 | 0.358 | 0.265 | 0.324 | 0.267 | 0.332 | 0.255 | 0.315 | 0.254 | 0.313 | 0.257 | 0.318 |
| ETTh1 | **0.399** | **0.417** | 0.439 | 0.448 | 0.419 | 0.436 | 0.417 | _0.426_ | 0.582 | 0.533 | 0.424 | 0.433 | 0.423 | 0.437 | 0.433 | 0.446 | _0.408_ | 0.427 | 0.427 | 0.441 |
| ETTh2 | 0.343 | 0.382 | 0.374 | 0.406 | 0.351 | 0.395 | 0.360 | 0.395 | 0.409 | 0.438 | 0.346 | 0.414 | 0.431 | 0.447 | _0.338_ | 0.393 | **0.335** | **0.386** | 0.347 | 0.394 |
| Traffic | _0.403_ | **0.257** | 0.428 | 0.282 | **0.397** | 0.275 | 0.416 | _0.264_ | 0.623 | 0.333 | 0.431 | 0.306 | 0.434 | 0.295 | 0.418 | 0.284 | 0.429 | 0.302 | 0.410 | 0.279 |
| Weather | **0.223** | **0.251** | 0.258 | 0.278 | _0.224_ | 0.261 | 0.225 | _0.258_ | 0.329 | 0.336 | 0.239 | 0.274 | 0.246 | 0.300 | 0.241 | 0.280 | 0.244 | 0.281 | 0.225 | 0.263 |
| Solar | **0.187** | **0.219** | 0.233 | 0.262 | 0.207 | 0.294 | 0.204 | _0.228_ | 0.233 | 0.290 | 0.233 | 0.290 | 0.230 | 0.295 | 0.235 | 0.269 | 0.232 | 0.268 | _0.203_ | 0.261 |
| Electricity | **0.158** | **0.250** | 0.178 | 0.270 | _0.159_ | _0.253_ | 0.168 | 0.261 | 0.195 | 0.296 | 0.164 | 0.259 | 0.166 | 0.264 | 0.164 | 0.259 | 0.169 | 0.265 | 0.185 | 0.284 |

Table 2: The average results of four prediction lengths in datasets injected with anomalies.

| Models | ours | | ITransformer | | PatchTST | | Pathformer | | TimesNet | | ModernTCN | | Dlinear | | TiDE | | FITS | | TimeMixer | |
|---|---|---|---|---|---|---|---|---|---|---|---|---|---|---|---|---|---|---|---|---|
| Metric | MSE | MAE | MSE | MAE | MSE | MAE | MSE | MAE | MSE | MAE | MSE | MAE | MSE | MAE | MSE | MAE | MSE | MAE | MSE | MAE |
| ETTm1 | **0.379** | **0.393** | 0.400 | 0.419 | 0.398 | 0.405 | 0.396 | 0.404 | 0.468 | 0.452 | 0.408 | 0.417 | _0.388_ | 0.405 | 0.391 | 0.405 | 0.389 | _0.403_ | 0.473 | 0.445 |
| ETTm2 | **0.270** | **0.320** | 0.307 | 0.353 | 0.289 | 0.337 | _0.281_ | _0.325_ | 0.337 | 0.365 | 0.307 | 0.357 | 0.332 | 0.388 | 0.289 | 0.344 | 0.288 | 0.342 | 0.293 | 0.347 |
| ETTh1 | **0.429** | **0.440** | 0.465 | 0.475 | 0.451 | 0.460 | 0.462 | 0.470 | 0.518 | 0.491 | 0.456 | _0.456_ | 0.441 | 0.458 | 0.450 | 0.462 | _0.440_ | 0.458 | 0.515 | 0.499 |
| ETTh2 | **0.345** | **0.386** | 0.396 | 0.426 | 0.366 | 0.396 | 0.356 | 0.399 | 0.448 | 0.447 | 0.370 | 0.412 | 0.560 | 0.524 | 0.356 | 0.401 | 0.382 | 0.422 | | |
| Traffic | **0.427** | **0.276** | 0.502 | 0.355 | _0.451_ | _0.290_ | 0.505 | 0.302 | 0.621 | 0.333 | 0.769 | 0.472 | 0.718 | 0.451 | 0.735 | 0.454 | 0.722 | 0.447 | 0.470 | 0.327 |
| Weather | **0.239** | **0.262** | 0.437 | 0.315 | _0.250_ | _0.281_ | 0.476 | 0.327 | 0.462 | 0.351 | 0.521 | 0.358 | 0.522 | 0.491 | 0.625 | 0.387 | 0.596 | 0.346 | 0.273 | 0.287 |
| Solar | **0.211** | **0.238** | 0.265 | 0.327 | 0.226 | 0.290 | 0.251 | 0.295 | 0.281 | 0.334 | 0.279 | 0.329 | 0.262 | 0.331 | 0.266 | 0.303 | 0.268 | 0.305 | _0.214_ | _0.273_ |
| Electricity | **0.175** | **0.263** | _0.186_ | _0.284_ | 0.226 | 0.296 | 0.217 | 0.291 | 0.314 | 0.394 | 0.247 | 0.344 | 0.255 | 0.355 | 0.268 | 0.355 | 0.270 | 0.355 | 0.296 | 0.382 |

## 5.2 MAIN RESULTS

**Results in Real-world Datasets.** Table 1 shows the prediction performance of different models on real-world datasets. RockTS leads on most datasets, achieving an average MSE reduction of 8%. RockTS exhibits significant advantages without injected anomalous subsequences, because sensor data inherently contains anomalies. Through end-to-end training, RockTS accurately detects such anomalies and converts them to normal patterns, reducing their interference and improving performance. Thus, RockTS adapts to a wide range of prediction scenarios.

**Result in Datasets Injected with Anomalous Subsequences.** We inject anomalous subsequences in datasets at a ratio of 10% to further evaluate models' robustness against anomalous subsequences. As shown in Table 2, RockTS outperforms all baselines across datasets injected with anomalous subsequences, achieving a MSE reduction of 21% in average. The full results are in Appendix H.

## 5.3 ABLATION STUDIES

**Detection Based on IB.** In Table 3, We performed two forms of ablation experiments on IB-based detction (IB-D): 1.replacing it with detecting randomly (Random-D) and 2.removing it which means that no regions are detected and the original series are directly used for prediction. We evaluate their effect on four datasets injuected with anomalous subsequences. The experimental results illustrate that our detection based on IB yields enhancements across all four datasets.

**Reconstruction Based on OT.** We also perform ablation studies for OT-based reconstruction (OT-R) by replacing it with simple reconstruction (R). Specifically, we remove the process of transport matrix-based adjustment, directly minimize the reconstruction error between $x$ and $\widetilde{x}$, and use $\widetilde{x}$ for the prediction task. Table 3 demonstrates that the model performance decreases when the OT-based reconstruction is removed, which proves that the OT-based reconstruction is effective in imputating the masked regions into normal patterns, helping to achieve more robust forecasting.

Table 3: Ablations on IB-based dectection and OT-based imputation.

| Design | | | | ETTh1 | | ETTm2 | | Electricity | | Traffic | |
|---|---|---|---|---|---|---|---|---|---|---|---|
| IB-D | Random-D | OT-R | R | MSE | MAE | MSE | MAE | MSE | MAE | MSE | MAE |
| ✓ | | ✓ | | 0.429 | 0.440 | 0.270 | 0.320 | 0.177 | 0.264 | 0.427 | 0.276 |
| | ✓ | ✓ | | 0.455 | 0.462 | 0.287 | 0.334 | 0.231 | 0.304 | 0.452 | 0.290 |
| ✓ | | | ✓ | 0.439 | 0.447 | 0.276 | 0.321 | 0.196 | 0.289 | 0.445 | 0.288 |
| | | | ✓ | 0.451 | 0.460 | 0.289 | 0.337 | 0.226 | 0.296 | 0.451 | 0.290 |

## 5.4 MODLE ANALYSIS

**Visualization.** We illustrate prediction showcases on two datasets with anomalous subsequences for RockTS and using only the predictor in Figure 4. Obviously, the framework of RockTS effectively improves the robustness of the model in the face of data containing anomalous subsequences, by effectively detecting and transforming anomalous patterns in the purple areas into normal patterns.

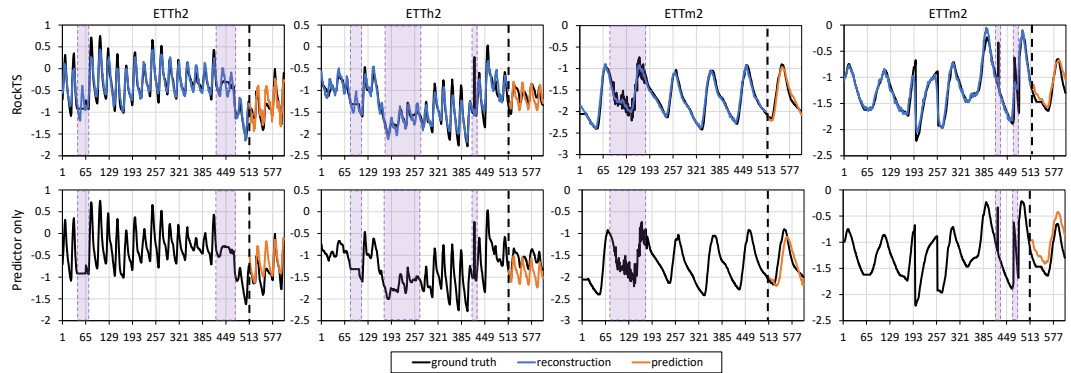

Figure 4: Prediction showcases of RockTS and predictor only.

**Anomalous Subsequences Ratio.** To explore the effect of the anomalous subsequences on different models, we inject six kinds of anomalous subsequences into ETTm2 and ETTh2 at the ratio from 2% to 20% with steps of 2%, respectively. Figure 5 illustrates the performance changes of different models and RockTS has significant advantages for different anomalous subsequences ratios.

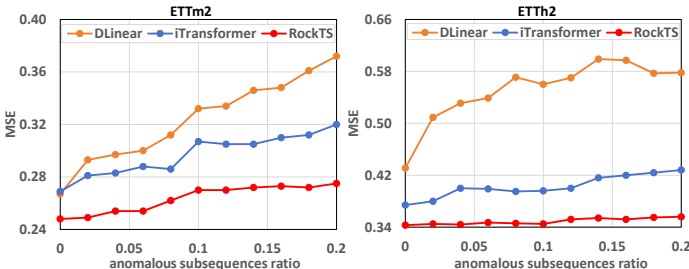

Figure 5: Models' performance when the ratio of anomalous subsequences gradually increases.

**Replacement of Predictors.** We port our detect-impute-forecast framework to mainstream forecasting models: iTransformer and DLinear. As shown in Table 4, our framework significantly improves the robust prediction capability of them. Specifically, in the case of DLinear on ETTh2, our framework enhances its prediction accuracy by 30%. Moreover, We also append the effects of an independent data cleaning prior to prediction that employs the widely-used Isolation Forest algorithm Liu et al. (2008) for detection and SAITS Du et al. (2023) for imputation. Although data cleaning provides some weak improvements in such scenarios, our framework improves the prediction effectiveness more significantly.

Table 4: Effect of porting our framework to different predictors, compared to data cleaning methods.

| Datasets | | ETTh2 | | ETTm2 | | Electricity | | Solar | |
|---|---|---|---|---|---|---|---|---|---|
| Metric | | MSE | MAE | MSE | MAE | MSE | MAE | MSE | MAE |
| Dlinear | base | 0.560 | 0.524 | 0.332 | 0.388 | 0.255 | 0.355 | 0.262 | 0.331 |
| | +ours | 0.361 | 0.398 | 0.314 | 0.377 | 0.242 | 0.349 | 0.256 | 0.313 |
| | +data clean | 0.551 | 0.517 | 0.321 | 0.382 | 0.250 | 0.351 | 0.259 | 0.326 |
| iTransformer | base | 0.396 | 0.426 | 0.307 | 0.353 | 0.186 | 0.284 | 0.265 | 0.327 |
| | +ours | 0.374 | 0.409 | 0.291 | 0.338 | 0.170 | 0.260 | 0.229 | 0.265 |
| | +data clean | 0.382 | 0.419 | 0.293 | 0.347 | 0.184 | 0.283 | 0.259 | 0.327 |

**More Experiments.** We show the sensitivity and efficiency analyses in Appendix D and E.

## 6 CONCLUSION

In this paper, we proposed RockTS, a novel end-to-end framework for robust time series forecasting that addresses the challenge of Anomalous subsequences in real-world data. RockTS integrates the detection and imputation of anomalous subsequences into forecasting through a unified optimization objective. RockTS shows superior performance in both real-world and synthetic data.

## 7 REPRODUCIBILITY STATEMENT

Reproducibility statement Our work meets reproducibility requirements. Specifically, you can obtain our code from the anonymous link: `https://anonymous.4open.science/r/RockTS-D08F`.

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

# A APPENDIX

# A IMPLEMENTATION DETAILS

## A.1 EVALUATION DATASETS

We use the following 8 multivariate time-series datasets for downstream forecasting task: ETT datasets[1] contain 7 variates collected from two different electric transformers from July 2016 to July 2018. It consists of four subsets, of which ETTh1/ETTh2 are recorded hourly and ETTm1/ETTm2 are recorded every 15 minutes. Electricity[2] contains the electricity consumption of 321 customers from July 2016 to July 2019, recorded hourly. Solar[3] collects production from 137 PV plants in Alabama, recorded every 10 minutes. Traffic[4] contains road occupancy rates measured by 862 sensors on freeways in the San Francisco Bay Area from 2015 to 2016, recorded hourly. Weather[5] collects 21 meteorological indicators, such as temperature and barometric pressure, for Germany in 2020, recorded every 10 minutes. We split each evaluation dataset into train-validation-test sets and detailed statistics of evaluation datasets are shown in Table 5.

Table 5: The statistics of evaluation datasets.

| Dataset | Domain | Frequency | Timestamps | Split | Dims |
|---|---|---|---|---|---|
| ETTh1 | Energy | 1 hour | 14400 | 6:2:2 | 7 |
| ETTh2 | Energy | 1 hour | 14400 | 6:2:2 | 7 |
| ETTm1 | Energy | 15 mins | 57600 | 6:2:2 | 7 |
| ETTm2 | Energy | 15 mins | 57600 | 6:2:2 | 7 |
| Electricity | Energy | 10 mins | 26304 | 7:1:2 | 321 |
| Solar | Energy | 10 mins | 52560 | 7:1:2 | 137 |
| Traffic | Traffic | 1 hour | 17544 | 7:1:2 | 862 |
| Weather | Environment | 10 mins | 52696 | 7:1:2 | 21 |

## A.2 ANOMALOUS SUBSEQUENCE INJECTION

To evaluate the robustness of the model on data that contains anomalous subsequences, we construct synthetic datasets by injecting six types of anomalous subsequences: vmirror, hmirror, scale, outlier, noise and pattern into real-world datasets. Specific examples of the injected anomalous subsequences are shown in Figure 6. The lookback windows use the time series injected with anomalous subsequences, while the prediction windows use the original time series as the ground truth.

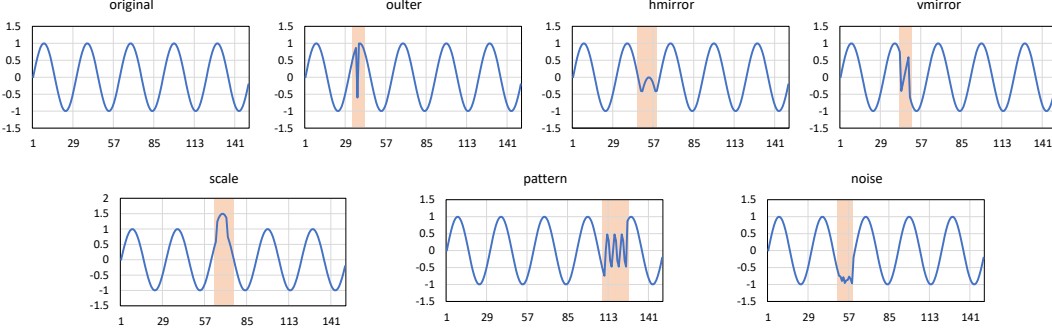

Figure 6: Anomalous subsequence injection. The blue lines represent the time series, while the orange areas intervals the anomalous subsequence we generated.

---

[1]https://github.com/zhouhaoyi/ETDataset

[2]https://archive.ics.uci.edu/ml/datasets/ElectricityLoadDiagrams20112014

[3]https://dl.acm.org/doi/abs/10.1145/3209978.3210006

[4]https://pems.dot.ca.gov/

[5]https://www.bgc-jena.mpg.de/wetter/

### A.3 BASELINES

We compare our model with nine state-of-the-art models for comprehensive evaluations, including Transformer-based models: iTransformer Liu et al. (2024), PatchTST Chen et al. (2024) and Path-Former Nie et al. (2023); CNN-based model: TimesNet Wu et al. (2022) and ModernTCN Luo & Wang (2024); MLP-based models: FITS Xu et al. (2024), TIDE Das et al. (2023), TimerMixer Wang et al. (2024) and DLinear Zeng et al. (2023). The specific code base for these models is listed in Table 6.

Table 6: Code repositories for baselines.

| Model Types | Models | Code Repositories |
|---|---|---|
| Transformer-based | iTransformer | https://github.com/thuml/iTransformer |
| | PatchTST | https://github.com/yuqinie98/PatchTST |
| | Pathformer | https://github.com/decisionintelligence/pathformer |
| CNN-based | TimesNet | https://github.com/thuml/TimesNet |
| | ModernTCN | https://github.com/luodhhh/ModernTCN |
| MLP-based | FITS | https://github.com/VEWOXIC/FITS |
| | TIDE | https://github.com/google-research/google-research/tree/master/tide |
| | TimeMixer | https://github.com/kwuking/TimeMixer |
| | DLinear | https://github.com/honeywell21/DLinear |

## B THEORETICAL ANALYSIS

In Section 7, we transformed the compactness term of the Information Bottleneck from mutual information minimization to its upper bound minimization form. In this section, we provide a detailed explanation of this derivation process. First, the mutual information $I(\mathbf{x}; \mathbf{x}_{\mathrm{m}})$ is defined as:

$$I(\mathbf{x}; \mathbf{x}_{\mathrm{m}}) = H(\mathbf{x}_{\mathrm{m}}) - H(\mathbf{x}_{\mathrm{m}}|\mathbf{x}) = \mathbb{E}_{\mathbf{x}, \mathbf{x}_{\mathrm{m}}}[\log \frac{\mathbb{P}(\mathbf{x}_{\mathrm{m}}|\mathbf{x})}{\mathbb{P}(\mathbf{x}_{\mathrm{m}})}] \tag{16}$$

Note that we introduced a trainable network $\mathbb{P}_\theta(\mathbf{x}_{\mathrm{m}}|\mathbf{x})$ to generate $\mathbf{x}_{\mathrm{m}}$. Meanwhile, since $\mathbb{P}(\mathbf{x}_{\mathrm{m}})$ is intractable, we leverage the non-negativity property of the Kullback-Leibler (KL) divergence to derive a variational approximation $\mathbb{Q}(\mathbf{x}_{\mathrm{m}})$ as a substitute for $\mathbb{P}(\mathbf{x}_{\mathrm{m}})$.

$$I(\mathbf{x}; \mathbf{x}_{\mathrm{m}}) = \mathbb{E}_{\mathbf{x}, \mathbf{x}_{\mathrm{m}}}[\log \frac{\mathbb{P}_\theta(\mathbf{x}_{\mathrm{m}}|\mathbf{x})}{\mathbb{P}(\mathbf{x}_{\mathrm{m}})}] \tag{17}$$

$$= \mathbb{E}_{\mathbf{x}, \mathbf{x}_{\mathrm{m}}}[\log \frac{\mathbb{P}_\theta(\mathbf{x}_{\mathrm{m}}|\mathbf{x})}{\mathbb{Q}(\mathbf{x}_{\mathrm{m}})}] + \mathbb{E}_{\mathbf{x}, \mathbf{x}_{\mathrm{m}}}[\log \frac{\mathbb{Q}(\mathbf{x}_{\mathrm{m}})}{\mathbb{P}(\mathbf{x}_{\mathrm{m}})}] \tag{18}$$

$$= \mathbb{E}_{\mathbf{x}, \mathbf{x}_{\mathrm{m}}}[\log \frac{\mathbb{P}_\theta(\mathbf{x}_{\mathrm{m}}|\mathbf{x})}{\mathbb{Q}(\mathbf{x}_{\mathrm{m}})}] + \mathbb{E}_{\mathbf{x}|\mathbf{x}_{\mathrm{m}}}[\mathbb{P}(\mathbf{x}_{\mathrm{m}}) \log \frac{\mathbb{Q}(\mathbf{x}_{\mathrm{m}})}{\mathbb{P}(\mathbf{x}_{\mathrm{m}})}] \tag{19}$$

$$= \mathbb{E}_{\mathbf{x}, \mathbf{x}_{\mathrm{m}}}[\log \frac{\mathbb{P}_\theta(\mathbf{x}_{\mathrm{m}}|\mathbf{x})}{\mathbb{Q}(\mathbf{x}_{\mathrm{m}})}] - \mathbb{E}_{\mathbf{x}|\mathbf{x}_{\mathrm{m}}}[D_{\mathrm{KL}}[\mathbb{P}(\mathbf{x}_{\mathrm{m}})||\mathbb{Q}(\mathbf{x}_{\mathrm{m}})]] \tag{20}$$

$$\geq \mathbb{E}_{\mathbf{x}, \mathbf{x}_{\mathrm{m}}}[\log \frac{\mathbb{P}_\theta(\mathbf{x}_{\mathrm{m}}|\mathbf{x})}{\mathbb{Q}(\mathbf{x}_{\mathrm{m}})}] \tag{21}$$

$$= \mathbb{E}_{\mathbf{x}}[D_{\mathrm{KL}}[\mathbb{P}_\theta(\mathbf{x}_{\mathrm{m}}|\mathbf{x})||\mathbb{Q}(\mathbf{x}_{\mathrm{m}})]], \tag{22}$$

Furthermore, we transform the problem of obtaining a subsequence $\mathbf{x}_{\mathrm{m}}$ into generating a stochastic mask $M$, where $\mathbf{x}_{\mathrm{m}} = \mathbf{x} \odot M$. Additionally, we define $\mathbb{Q}(M)$ as a Bernoulli distribution with a sparsity parameter $\tau$ to control the mask generation. Through this transformation, the original compactness constraint term is converted into a more tractable loss function:

$$\mathbb{E}_{\mathbf{x}}[D_{\mathrm{KL}}[\mathbb{P}_\theta(\mathbf{M}|\mathbf{x})||\mathbb{Q}(\mathbf{M})]] = \sum_{i=1}^{L} \left[ \lambda_i \log \left( \frac{\lambda_i}{\tau} \right) + (1 - \lambda_i) \log \left( \frac{1 - \lambda_i}{1 - \tau} \right) \right] \tag{23}$$

## C MORE COMPARISION

In this section, we compare RockTS with other robust time series forecasting methods. However, existing works in this direction either do not provide open-source codes or follow diverse experimental settings, making it hard for a unified comparison. Thus, we select a recently proposed robust

time series forecasting methods, TAFAS Kim et al. (2025), for comparison to further illustrate the superior robustness of RockTS. Since TAFAS do not have open-source code, we report results from its original paper and test RockTS in the same experimental settings as it to ensure a fair comparison. Specifically, we keep the input length to 96. The results are shown in Table 7. When compared to the robust time series forecasting method, RockTS still achieves the lowest average MSE and MAE on most datasets.

Table 7: Comparison results with robust time series forecasting TAFAS.

| Models | | ETTm1 | | ETTm2 | | ETTh1 | | ETTh2 | |
|---|---|---|---|---|---|---|---|---|---|
| Metric | | MSE | MAE | MSE | MAE | MSE | MAE | MSE | MAE |
| RockTS | 96 | **0.364** | **0.375** | **0.155** | **0.253** | 0.441 | **0.439** | **0.225** | **0.312** |
| | 192 | **0.427** | **0.412** | **0.189** | **0.283** | 0.494 | **0.477** | **0.276** | **0.348** |
| | 336 | **0.485** | **0.455** | **0.228** | **0.313** | 0.543 | **0.510** | **0.309** | **0.372** |
| | 720 | **0.536** | **0.503** | **0.298** | **0.358** | 0.688 | **0.602** | **0.385** | **0.418** |
| | avg | **0.453** | **0.436** | **0.218** | **0.302** | 0.541 | **0.507** | **0.299** | **0.363** |
| TSFAS | 96 | 0.377 | 0.397 | 0.156 | 0.262 | **0.429** | 0.444 | 0.232 | 0.320 |
| | 192 | 0.429 | 0.428 | 0.194 | 0.294 | **0.481** | 0.483 | 0.277 | 0.353 |
| | 336 | 0.487 | 0.461 | 0.232 | 0.323 | **0.529** | 0.519 | 0.318 | 0.382 |
| | 720 | 0.542 | 0.509 | 0.299 | 0.367 | 0.690 | 0.621 | 0.396 | 0.427 |
| | avg | 0.459 | 0.449 | 0.220 | 0.312 | **0.532** | 0.517 | 0.306 | 0.371 |

## D    SENSITIVITY

**Multi-loss Balance.** We use hyper-parameters $\alpha$ and $\beta$ to balance the loss functions, and set them to 1 by default. We performed a sensitivity analysis for them on four datasets, as shown in Figure 7. As $\alpha$ and $\beta$ changes, RockTS's performance shows only Minimal changes in average MSE. From the experimental results, it is clear that RockTS is not sensitive to the weights of the loss. Therefore, balancing the multi-loss during training of the model is not difficult.

**Sparsity of the Masks $\tau$.** $\tau$ is a significant parameter in training RockTS, which controls the sparsity of the masks and is set to 0.9 by default. We performed a sensitivity analysis to scrutinize the impact of varying $\tau$ on the forecasting performance. Figure 7 illustrates the relationship between the effect of RockTS and the sparsity parameter $\tau$ on four datasets. It is worth noting that RockTS's performance remains stable when $\tau$ within the range of 0.7 to 0.95, suggesting that the effectiveness of the interpreter is relatively insensitive to the selection of $\tau$ in this interval.

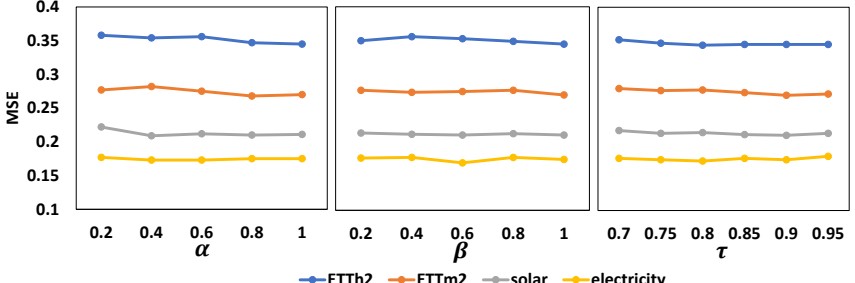

Figure 7: Sensitivity analyses on weights of the loss and sparsity of the masks on four datasets injected with anomalous subsequences.

## E    EFFICIENCY

As in Table 8, RockTS takes longer about 1.3 times for tranining and 2.0–2.4 times for inference than PatchTST (predictor only) with batch size of 1 and prediction length of 96, we also give the confidence interval of 95%. The additional cost for the improved accuracy is reasonable, considering the anomalous subsequences detection and the imputation. Due to the inference time per sample is only 9.8–12.3 ms, we believe that RockTS can also be deployed in a real-time environment.

Table 8: Comparison of efficiency with PatchTST on four real-world datasets.

| Phase | Model | ETTh1 | Weather | Electricity | Traffic |
|---|---|---|---|---|---|
| Train (sec/iter) | RockTS | 0.0098 ± 0.0003 | 0.0102 ± 0.0003 | 0.0108 ± 0.0003 | 0.0123 ± 0.0004 |
| | PatchTST | 0.0074 ± 0.0002 | 0.0078 ± 0.0002 | 0.0086 ± 0.0003 | 0.0091 ± 0.0002 |
| Inference (sec/iter) | RockTS | 0.0074 ± 0.0002 | 0.0074 ± 0.0002 | 0.0081 ± 0.0002 | 0.0087 ± 0.0002 |
| | PatchTST | 0.0031 ± 0.0001 | 0.0036 ± 0.0001 | 0.0038 ± 0.0001 | 0.0040± 0.0001 |

## F RESULTS DEVIATION

We have conducted RockTS three times with different random seeds and have recorded the standard deviations, as illustrated in Table 9. It can be observed that RockTS exhibits stable performance.

Table 9: Results deviation.

| Models | RockTS for real-word datasets | | RockTS for synthetic datasets | | confidence interval |
|---|---|---|---|---|---|
| Metric | MSE | MAE | MSE | MAE | - |
| ETTm1 | 0.345±0.003 | 0.368±0.002 | 0.379±0.004 | 0.393±0.005 | |
| ETTm2 | 0.248±0.003 | 0.303±0.003 | 0.27±0.006 | 0.32±0.006 | |
| ETTh1 | 0.399±0.002 | 0.417±0.003 | 0.429±0.004 | 0.44±0.003 | |
| ETTh2 | 0.343±0.004 | 0.382±0.002 | 0.345±0.004 | 0.386±0.004 | |
| Traffic | 0.403±0.005 | 0.257±0.003 | 0.427±0.007 | 0.276±0.006 | 99% |
| Weather | 0.223±0.005 | 0.251±0.005 | 0.239±0.012 | 0.262±0.010 | |
| Solar | 0.187±0.003 | 0.219±0.003 | 0.211±0.005 | 0.238±0.008 | |
| Electricity | 0.158±0.006 | 0.25±0.008 | 0.175±0.008 | 0.263±0.010 | |

## G BROADER IMPACTS

RockTS has potential positive social impacts in critical domains that rely on robust time series forecasting, such as energy, transportation, and environmental monitoring. By mitigating the effects of anomalous data, RockTS could improve prediction accuracy and robustness, as well as decision making for management.

# H   FULL RESULTS

In this section, we provide full results of RockTS and the compared baselines in the experiments on real-world datasets with prediction length 96 and batch size of 16, synthetic datasets and ablation studies.

Table 10: Full results on real-world datasets.

| Models | | RockTS | | ITransformer | | PatchTST | | Pathformer | | TimesNet | | ModernTCN | | Dlinear | | TiDE | | FITS | | TimeMixer | |
|---|---|---|---|---|---|---|---|---|---|---|---|---|---|---|---|---|---|---|---|---|---|
| Metric | | MSE | MAE | MSE | MAE | MSE | MAE | MSE | MAE | MSE | MAE | MSE | MAE | MSE | MAE | MSE | MAE | MSE | MAE | MSE | MAE |
| ETTm1 | 96 | 0.277 | 0.325 | 0.300 | 0.353 | 0.289 | 0.343 | 0.290 | 0.335 | 0.405 | 0.421 | 0.309 | 0.355 | 0.299 | 0.343 | 0.308 | 0.350 | 0.303 | 0.345 | 0.293 | 0.345 |
| | 192 | 0.327 | 0.356 | 0.345 | 0.382 | 0.329 | 0.368 | 0.337 | 0.363 | 0.508 | 0.473 | 0.342 | 0.365 | 0.335 | 0.365 | 0.338 | 0.367 | 0.337 | 0.365 | 0.335 | 0.372 |
| | 336 | 0.362 | 0.378 | 0.374 | 0.398 | 0.362 | 0.390 | 0.374 | 0.384 | 0.523 | 0.479 | 0.372 | 0.392 | 0.369 | 0.386 | 0.366 | 0.386 | 0.368 | 0.384 | 0.368 | 0.386 |
| | 720 | 0.416 | 0.412 | 0.429 | 0.430 | 0.416 | 0.423 | 0.428 | 0.416 | 0.523 | 0.484 | 0.421 | 0.418 | 0.425 | 0.421 | 0.426 | 0.419 | 0.420 | 0.413 | 0.426 | 0.417 |
| | avg | 0.345 | 0.368 | 0.362 | 0.391 | 0.349 | 0.381 | 0.357 | 0.375 | 0.490 | 0.464 | 0.361 | 0.430 | 0.357 | 0.379 | 0.360 | 0.381 | 0.357 | 0.377 | 0.356 | 0.380 |
| ETTm2 | 96 | 0.159 | 0.243 | 0.175 | 0.266 | 0.165 | 0.255 | 0.164 | 0.250 | 0.233 | 0.305 | 0.171 | 0.262 | 0.167 | 0.260 | 0.166 | 0.256 | 0.165 | 0.254 | 0.165 | 0.256 |
| | 192 | 0.216 | 0.282 | 0.242 | 0.312 | 0.221 | 0.293 | 0.219 | 0.288 | 0.265 | 0.328 | 0.230 | 0.304 | 0.224 | 0.303 | 0.221 | 0.293 | 0.219 | 0.291 | 0.225 | 0.298 |
| | 336 | 0.266 | 0.317 | 0.282 | 0.340 | 0.276 | 0.327 | 0.267 | 0.319 | 0.379 | 0.392 | 0.277 | 0.332 | 0.281 | 0.342 | 0.275 | 0.329 | 0.272 | 0.326 | 0.277 | 0.332 |
| | 720 | 0.352 | 0.372 | 0.378 | 0.398 | 0.362 | 0.381 | 0.361 | 0.377 | 0.390 | 0.407 | 0.381 | 0.398 | 0.397 | 0.421 | 0.361 | 0.382 | 0.359 | 0.381 | 0.360 | 0.387 |
| | avg | 0.248 | 0.303 | 0.269 | 0.329 | 0.256 | 0.314 | 0.253 | 0.309 | 0.317 | 0.358 | 0.265 | 0.324 | 0.267 | 0.332 | 0.255 | 0.315 | 0.254 | 0.313 | 0.257 | 0.318 |
| ETTh1 | 96 | 0.361 | 0.386 | 0.386 | 0.405 | 0.377 | 0.397 | 0.372 | 0.392 | 0.470 | 0.470 | 0.377 | 0.402 | 0.375 | 0.399 | 0.393 | 0.418 | 0.376 | 0.396 | 0.376 | 0.401 |
| | 192 | 0.397 | 0.411 | 0.424 | 0.440 | 0.409 | 0.425 | 0.408 | 0.415 | 0.568 | 0.523 | 0.415 | 0.421 | 0.405 | 0.416 | 0.433 | 0.442 | 0.400 | 0.418 | 0.413 | 0.430 |
| | 336 | 0.415 | 0.425 | 0.449 | 0.460 | 0.431 | 0.444 | 0.438 | 0.434 | 0.595 | 0.547 | 0.437 | 0.434 | 0.439 | 0.443 | 0.426 | 0.442 | 0.419 | 0.435 | 0.438 | 0.450 |
| | 720 | 0.421 | 0.447 | 0.495 | 0.487 | 0.457 | 0.477 | 0.450 | 0.463 | 0.694 | 0.591 | 0.468 | 0.473 | 0.472 | 0.490 | 0.478 | 0.484 | 0.435 | 0.458 | 0.483 | 0.483 |
| | avg | 0.399 | 0.417 | 0.439 | 0.448 | 0.419 | 0.436 | 0.417 | 0.426 | 0.582 | 0.533 | 0.424 | 0.433 | 0.423 | 0.437 | 0.433 | 0.446 | 0.408 | 0.427 | 0.427 | 0.441 |
| ETTh2 | 96 | 0.274 | 0.330 | 0.297 | 0.348 | 0.274 | 0.337 | 0.279 | 0.336 | 0.351 | 0.399 | 0.278 | 0.424 | 0.289 | 0.353 | 0.282 | 0.352 | 0.277 | 0.345 | 0.270 | 0.342 |
| | 192 | 0.341 | 0.372 | 0.371 | 0.403 | 0.348 | 0.384 | 0.345 | 0.380 | 0.394 | 0.429 | 0.343 | 0.388 | 0.383 | 0.418 | 0.334 | 0.387 | 0.331 | 0.379 | 0.349 | 0.387 |
| | 336 | 0.366 | 0.400 | 0.404 | 0.428 | 0.377 | 0.416 | 0.378 | 0.408 | 0.415 | 0.443 | 0.357 | 0.405 | 0.448 | 0.465 | 0.329 | 0.389 | 0.350 | 0.396 | 0.367 | 0.410 |
| | 720 | 0.391 | 0.426 | 0.424 | 0.444 | 0.406 | 0.441 | 0.437 | 0.455 | 0.477 | 0.481 | 0.406 | 0.438 | 0.605 | 0.551 | 0.405 | 0.445 | 0.382 | 0.425 | 0.401 | 0.436 |
| | avg | 0.343 | 0.382 | 0.374 | 0.406 | 0.351 | 0.395 | 0.360 | 0.395 | 0.409 | 0.438 | 0.346 | 0.414 | 0.431 | 0.447 | 0.338 | 0.393 | 0.335 | 0.386 | 0.347 | 0.394 |
| Traffic | 96 | 0.377 | 0.245 | 0.395 | 0.268 | 0.370 | 0.262 | 0.384 | 0.250 | 0.611 | 0.323 | 0.406 | 0.294 | 0.410 | 0.282 | 0.395 | 0.272 | 0.400 | 0.280 | 0.369 | 0.256 |
| | 192 | 0.394 | 0.252 | 0.417 | 0.276 | 0.386 | 0.269 | 0.405 | 0.257 | 0.609 | 0.317 | 0.417 | 0.298 | 0.423 | 0.287 | 0.402 | 0.273 | 0.412 | 0.288 | 0.400 | 0.271 |
| | 336 | 0.403 | 0.257 | 0.433 | 0.283 | 0.396 | 0.275 | 0.424 | 0.265 | 0.616 | 0.335 | 0.427 | 0.305 | 0.436 | 0.296 | 0.416 | 0.282 | 0.426 | 0.301 | 0.407 | 0.272 |
| | 720 | 0.438 | 0.276 | 0.467 | 0.302 | 0.435 | 0.295 | 0.452 | 0.283 | 0.656 | 0.349 | 0.473 | 0.327 | 0.466 | 0.315 | 0.457 | 0.309 | 0.478 | 0.339 | 0.462 | 0.316 |
| | avg | 0.403 | 0.257 | 0.428 | 0.282 | 0.397 | 0.275 | 0.416 | 0.264 | 0.623 | 0.333 | 0.431 | 0.306 | 0.434 | 0.295 | 0.418 | 0.284 | 0.429 | 0.302 | 0.410 | 0.279 |
| Weather | 96 | 0.147 | 0.185 | 0.174 | 0.214 | 0.149 | 0.196 | 0.148 | 0.195 | 0.193 | 0.244 | 0.149 | 0.204 | 0.176 | 0.237 | 0.173 | 0.225 | 0.172 | 0.225 | 0.147 | 0.198 |
| | 192 | 0.188 | 0.227 | 0.221 | 0.254 | 0.191 | 0.239 | 0.191 | 0.235 | 0.320 | 0.329 | 0.201 | 0.249 | 0.220 | 0.282 | 0.217 | 0.262 | 0.215 | 0.261 | 0.191 | 0.242 |
| | 336 | 0.240 | 0.269 | 0.278 | 0.296 | 0.242 | 0.279 | 0.243 | 0.274 | 0.363 | 0.366 | 0.257 | 0.291 | 0.265 | 0.319 | 0.253 | 0.293 | 0.261 | 0.295 | 0.244 | 0.280 |
| | 720 | 0.316 | 0.322 | 0.358 | 0.347 | 0.312 | 0.330 | 0.318 | 0.326 | 0.440 | 0.404 | 0.347 | 0.350 | 0.323 | 0.362 | 0.324 | 0.340 | 0.326 | 0.341 | 0.320 | 0.331 |
| | avg | 0.223 | 0.251 | 0.258 | 0.278 | 0.224 | 0.261 | 0.225 | 0.258 | 0.329 | 0.336 | 0.239 | 0.274 | 0.246 | 0.300 | 0.241 | 0.280 | 0.244 | 0.281 | 0.225 | 0.263 |
| Solar | 96 | 0.172 | 0.209 | 0.203 | 0.237 | 0.190 | 0.273 | 0.218 | 0.235 | 0.221 | 0.277 | 0.198 | 0.275 | 0.206 | 0.281 | 0.210 | 0.260 | 0.208 | 0.255 | 0.180 | 0.233 |
| | 192 | 0.184 | 0.219 | 0.233 | 0.261 | 0.204 | 0.302 | 0.196 | 0.220 | 0.215 | 0.280 | 0.201 | 0.282 | 0.225 | 0.291 | 0.231 | 0.270 | 0.229 | 0.267 | 0.201 | 0.259 |
| | 336 | 0.192 | 0.223 | 0.248 | 0.273 | 0.212 | 0.293 | 0.195 | 0.220 | 0.266 | 0.314 | 0.213 | 0.290 | 0.240 | 0.300 | 0.246 | 0.272 | 0.241 | 0.273 | 0.214 | 0.272 |
| | 720 | 0.199 | 0.227 | 0.249 | 0.275 | 0.221 | 0.310 | 0.208 | 0.237 | 0.231 | 0.291 | 0.255 | 0.289 | 0.248 | 0.307 | 0.252 | 0.273 | 0.248 | 0.277 | 0.218 | 0.278 |
| | avg | 0.187 | 0.219 | 0.233 | 0.262 | 0.207 | 0.294 | 0.204 | 0.228 | 0.233 | 0.290 | 0.233 | 0.290 | 0.230 | 0.295 | 0.235 | 0.269 | 0.232 | 0.268 | 0.203 | 0.261 |
| Electricity | 96 | 0.129 | 0.222 | 0.148 | 0.240 | 0.129 | 0.222 | 0.135 | 0.222 | 0.182 | 0.287 | 0.133 | 0.228 | 0.140 | 0.237 | 0.141 | 0.237 | 0.139 | 0.237 | 0.153 | 0.256 |
| | 192 | 0.146 | 0.238 | 0.162 | 0.253 | 0.147 | 0.240 | 0.157 | 0.253 | 0.193 | 0.293 | 0.146 | 0.241 | 0.153 | 0.249 | 0.147 | 0.244 | 0.154 | 0.250 | 0.168 | 0.269 |
| | 336 | 0.162 | 0.254 | 0.178 | 0.269 | 0.163 | 0.259 | 0.170 | 0.267 | 0.196 | 0.298 | 0.162 | 0.259 | 0.169 | 0.267 | 0.165 | 0.261 | 0.170 | 0.268 | 0.189 | 0.291 |
| | 720 | 0.197 | 0.285 | 0.225 | 0.317 | 0.197 | 0.290 | 0.211 | 0.302 | 0.209 | 0.307 | 0.214 | 0.307 | 0.203 | 0.301 | 0.204 | 0.292 | 0.212 | 0.304 | 0.228 | 0.320 |
| | avg | 0.158 | 0.250 | 0.178 | 0.270 | 0.159 | 0.253 | 0.168 | 0.261 | 0.195 | 0.296 | 0.164 | 0.259 | 0.166 | 0.264 | 0.164 | 0.259 | 0.169 | 0.265 | 0.185 | 0.284 |

Table 11: Full results on datasets injected with anomalous subsequences.

| Models | | RockTS | | ITransformer | | PatchTST | | Pathformer | | TimesNet | | ModernTCN | | Dlinear | | TiDE | | FITS | | TimeMixer | |
|---|---|---|---|---|---|---|---|---|---|---|---|---|---|---|---|---|---|---|---|---|---|
| Metric | | MSE | MAE | MSE | MAE | MSE | MAE | MSE | MAE | MSE | MAE | MSE | MAE | MSE | MAE | MSE | MAE | MSE | MAE | MSE | MAE |
| ETTm1 | 96 | 0.320 | 0.354 | 0.342 | 0.386 | 0.354 | 0.374 | 0.336 | 0.370 | 0.401 | 0.416 | 0.354 | 0.387 | 0.338 | 0.376 | 0.342 | 0.377 | 0.339 | 0.375 | 0.402 | 0.404 |
| | 192 | 0.359 | 0.383 | 0.379 | 0.406 | 0.375 | 0.395 | 0.373 | 0.393 | 0.459 | 0.444 | 0.385 | 0.405 | 0.366 | 0.392 | 0.368 | 0.392 | 0.368 | 0.391 | 0.445 | 0.429 |
| | 336 | 0.389 | 0.400 | 0.410 | 0.425 | 0.407 | 0.410 | 0.408 | 0.409 | 0.483 | 0.463 | 0.427 | 0.429 | 0.396 | 0.410 | 0.396 | 0.410 | 0.397 | 0.408 | 0.477 | 0.448 |
| | 720 | 0.449 | 0.432 | 0.471 | 0.457 | 0.456 | 0.440 | 0.467 | 0.443 | 0.530 | 0.487 | 0.468 | 0.449 | 0.452 | 0.442 | 0.458 | 0.442 | 0.454 | 0.438 | 0.566 | 0.497 |
| | avg | 0.379 | 0.393 | 0.400 | 0.419 | 0.398 | 0.405 | 0.396 | 0.404 | 0.468 | 0.452 | 0.408 | 0.417 | 0.388 | 0.405 | 0.391 | 0.405 | 0.389 | 0.403 | 0.473 | 0.445 |
| ETTm2 | 96 | 0.188 | 0.264 | 0.214 | 0.294 | 0.200 | 0.279 | 0.205 | 0.273 | 0.232 | 0.304 | 0.220 | 0.304 | 0.218 | 0.312 | 0.208 | 0.293 | 0.206 | 0.291 | 0.225 | 0.313 |
| | 192 | 0.239 | 0.301 | 0.284 | 0.336 | 0.257 | 0.318 | 0.251 | 0.306 | 0.295 | 0.343 | 0.271 | 0.335 | 0.275 | 0.354 | 0.256 | 0.323 | 0.254 | 0.322 | 0.248 | 0.316 |
| | 336 | 0.285 | 0.331 | 0.329 | 0.368 | 0.318 | 0.353 | 0.294 | 0.336 | 0.353 | 0.376 | 0.328 | 0.367 | 0.346 | 0.401 | 0.305 | 0.353 | 0.303 | 0.353 | 0.303 | 0.352 |
| | 720 | 0.367 | 0.382 | 0.400 | 0.412 | 0.381 | 0.396 | 0.373 | 0.383 | 0.469 | 0.436 | 0.411 | 0.424 | 0.489 | 0.485 | 0.386 | 0.404 | 0.387 | 0.404 | 0.395 | 0.408 |
| | avg | 0.270 | 0.320 | 0.307 | 0.353 | 0.289 | 0.337 | 0.281 | 0.325 | 0.337 | 0.365 | 0.307 | 0.357 | 0.332 | 0.388 | 0.289 | 0.344 | 0.288 | 0.342 | 0.293 | 0.347 |
| ETTh1 | 96 | 0.390 | 0.409 | 0.417 | 0.442 | 0.409 | 0.431 | 0.422 | 0.438 | 0.451 | 0.456 | 0.412 | 0.428 | 0.394 | 0.420 | 0.415 | 0.437 | 0.395 | 0.421 | 0.425 | 0.445 |
| | 192 | 0.430 | 0.434 | 0.448 | 0.460 | 0.458 | 0.460 | 0.468 | 0.466 | 0.508 | 0.487 | 0.450 | 0.447 | 0.447 | 0.461 | 0.450 | 0.471 | 0.426 | 0.443 | 0.453 | 0.462 |
| | 336 | 0.443 | 0.448 | 0.465 | 0.474 | 0.471 | 0.470 | 0.472 | 0.484 | 0.549 | 0.503 | 0.466 | 0.457 | 0.448 | 0.454 | 0.443 | 0.458 | 0.470 | 0.479 | 0.471 | 0.479 |
| | 720 | 0.451 | 0.470 | 0.529 | 0.526 | 0.465 | 0.477 | 0.486 | 0.495 | 0.562 | 0.520 | 0.494 | 0.491 | 0.475 | 0.497 | 0.492 | 0.496 | 0.471 | 0.488 | 0.710 | 0.611 |
| | avg | 0.429 | 0.440 | 0.465 | 0.475 | 0.451 | 0.460 | 0.462 | 0.470 | 0.518 | 0.491 | 0.456 | 0.456 | 0.441 | 0.458 | 0.450 | 0.462 | 0.440 | 0.458 | 0.515 | 0.499 |
| ETTh2 | 96 | 0.293 | 0.349 | 0.325 | 0.379 | 0.306 | 0.358 | 0.302 | 0.359 | 0.376 | 0.402 | 0.316 | 0.373 | 0.360 | 0.419 | 0.301 | 0.367 | 0.300 | 0.362 | 0.343 | 0.390 |
| | 192 | 0.343 | 0.382 | 0.398 | 0.422 | 0.363 | 0.395 | 0.357 | 0.398 | 0.436 | 0.437 | 0.361 | 0.405 | 0.445 | 0.473 | 0.343 | 0.395 | 0.346 | 0.393 | 0.395 | 0.395 |
| | 336 | 0.357 | 0.394 | 0.438 | 0.449 | 0.367 | 0.404 | 0.377 | 0.400 | 0.466 | 0.462 | 0.367 | 0.415 | 0.541 | 0.525 | 0.363 | 0.394 | 0.359 | 0.407 | 0.363 | 0.419 |
| | 720 | 0.386 | 0.422 | 0.424 | 0.454 | 0.428 | 0.428 | 0.387 | 0.440 | 0.514 | 0.487 | 0.438 | 0.456 | 0.894 | 0.679 | 0.416 | 0.450 | 0.402 | 0.438 | 0.402 | 0.483 |
| | avg | 0.345 | 0.386 | 0.396 | 0.426 | 0.366 | 0.396 | 0.356 | 0.399 | 0.448 | 0.447 | 0.370 | 0.412 | 0.560 | 0.524 | 0.356 | 0.401 | 0.352 | 0.400 | 0.382 | 0.422 |
| Traffic | 96 | 0.404 | 0.263 | 0.465 | 0.338 | 0.427 | 0.275 | 0.420 | 0.290 | 0.598 | 0.321 | 0.760 | 0.478 | 0.697 | 0.445 | 0.706 | 0.445 | 0.698 | 0.442 | 0.448 | 0.327 |
| | 192 | 0.418 | 0.271 | 0.476 | 0.346 | 0.441 | 0.286 | 0.512 | 0.292 | 0.614 | 0.325 | 0.746 | 0.467 | 0.704 | 0.446 | 0.716 | 0.447 | 0.707 | 0.443 | 0.446 | 0.346 |
| | 336 | 0.426 | 0.274 | 0.505 | 0.356 | 0.447 | 0.288 | 0.526 | 0.304 | 0.614 | 0.330 | 0.769 | 0.464 | 0.716 | 0.450 | 0.736 | 0.453 | 0.720 | 0.445 | 0.447 | 0.319 |
| | 720 | 0.461 | 0.293 | 0.560 | 0.380 | 0.489 | 0.313 | 0.562 | 0.321 | 0.657 | 0.354 | 0.801 | 0.477 | 0.753 | 0.462 | 0.783 | 0.470 | 0.763 | 0.458 | 0.470 | 0.316 |
| | avg | 0.427 | 0.276 | 0.502 | 0.355 | 0.451 | 0.290 | 0.505 | 0.302 | 0.621 | 0.333 | 0.769 | 0.472 | 0.718 | 0.451 | 0.735 | 0.454 | 0.722 | 0.447 | 0.470 | 0.327 |
| Weather | 96 | 0.167 | 0.200 | 0.391 | 0.272 | 0.170 | 0.218 | 0.380 | 0.267 | 0.401 | 0.305 | 0.487 | 0.330 | 0.494 | 0.470 | 0.576 | 0.357 | 0.474 | 0.290 | 0.173 | 0.216 |
| | 192 | 0.206 | 0.238 | 0.382 | 0.290 | 0.215 | 0.260 | 0.440 | 0.297 | 0.423 | 0.333 | 0.490 | 0.337 | 0.506 | 0.478 | 0.607 | 0.374 | 0.566 | 0.328 | 0.218 | 0.254 |
| | 336 | 0.256 | 0.277 | 0.439 | 0.326 | 0.280 | 0.303 | 0.503 | 0.357 | 0.480 | 0.364 | 0.561 | 0.363 | 0.530 | 0.503 | 0.624 | 0.390 | 0.592 | 0.355 | 0.269 | 0.294 |
| | 720 | 0.327 | 0.330 | 0.535 | 0.370 | 0.335 | 0.344 | 0.582 | 0.386 | 0.546 | 0.404 | 0.546 | 0.404 | 0.558 | 0.516 | 0.693 | 0.428 | 0.751 | 0.410 | 0.433 | 0.383 |
| | avg | 0.239 | 0.262 | 0.437 | 0.315 | 0.250 | 0.281 | 0.476 | 0.327 | 0.462 | 0.352 | 0.521 | 0.358 | 0.522 | 0.491 | 0.625 | 0.387 | 0.596 | 0.346 | 0.273 | 0.287 |
| Solar | 96 | 0.200 | 0.230 | 0.240 | 0.316 | 0.213 | 0.280 | 0.226 | 0.254 | 0.256 | 0.317 | 0.259 | 0.313 | 0.244 | 0.318 | 0.247 | 0.292 | 0.249 | 0.295 | 0.201 | 0.262 |
| | 192 | 0.210 | 0.239 | 0.268 | 0.322 | 0.226 | 0.290 | 0.248 | 0.299 | 0.291 | 0.344 | 0.276 | 0.327 | 0.260 | 0.330 | 0.262 | 0.301 | 0.267 | 0.305 | 0.210 | 0.270 |
| | 336 | 0.214 | 0.240 | 0.278 | 0.336 | 0.232 | 0.298 | 0.261 | 0.306 | 0.293 | 0.359 | 0.286 | 0.333 | 0.269 | 0.336 | 0.274 | 0.308 | 0.276 | 0.310 | 0.219 | 0.283 |
| | 720 | 0.219 | 0.242 | 0.272 | 0.333 | 0.233 | 0.293 | 0.269 | 0.323 | 0.284 | 0.317 | 0.295 | 0.343 | 0.274 | 0.341 | 0.279 | 0.310 | 0.279 | 0.310 | 0.225 | 0.278 |
| | avg | 0.211 | 0.238 | 0.265 | 0.327 | 0.226 | 0.290 | 0.251 | 0.295 | 0.281 | 0.334 | 0.279 | 0.329 | 0.262 | 0.331 | 0.266 | 0.302 | 0.268 | 0.305 | 0.214 | 0.273 |
| Electricity | 96 | 0.146 | 0.237 | 0.158 | 0.259 | 0.233 | 0.283 | 0.186 | 0.263 | 0.284 | 0.373 | 0.224 | 0.326 | 0.238 | 0.340 | 0.246 | 0.337 | 0.246 | 0.337 | 0.319 | 0.401 |
| | 192 | 0.162 | 0.251 | 0.174 | 0.274 | 0.211 | 0.281 | 0.202 | 0.274 | 0.312 | 0.395 | 0.233 | 0.333 | 0.245 | 0.346 | 0.255 | 0.345 | 0.256 | 0.345 | 0.255 | 0.352 |
| | 336 | 0.178 | 0.266 | 0.189 | 0.289 | 0.219 | 0.299 | 0.218 | 0.293 | 0.329 | 0.400 | 0.249 | 0.348 | 0.255 | 0.356 | 0.268 | 0.354 | 0.270 | 0.356 | 0.255 | 0.347 |
| | 720 | 0.214 | 0.297 | 0.221 | 0.315 | 0.242 | 0.319 | 0.263 | 0.333 | 0.331 | 0.409 | 0.280 | 0.370 | 0.282 | 0.378 | 0.305 | 0.382 | 0.306 | 0.382 | 0.357 | 0.426 |
| | avg | 0.175 | 0.263 | 0.186 | 0.284 | 0.226 | 0.296 | 0.217 | 0.291 | 0.314 | 0.394 | 0.247 | 0.344 | 0.255 | 0.355 | 0.268 | 0.355 | 0.270 | 0.355 | 0.296 | 0.382 |

