# OpenReview forum: "RockTS: Robust Time Series Forecasting based on Information Bottleneck and Optimal Transport"
_ICLR.cc/2026/Conference — Submitted to ICLR 2026_

### Official Review · Reviewer_iSu2 · 2025-10-27

**Soundness:** 3
**Presentation:** 3
**Contribution:** 3
**Rating:** 6
**Confidence:** 4

**Summary:**

This paper presents RockTS, a novel, end-to-end framework designed to tackle the critical real-world problem of robust time series forecasting by jointly addressing data anomalies. The core contribution is an integrated pipeline that utilizes the Information Bottleneck principle to effectively detect anomalous subsequences and subsequently employs Optimal Transport to perform principled and robust imputation, demonstrating significant performance gains over traditional forecasting methods when dealing with noisy and contaminated datasets.

**Strengths:**

1. The paper's motivation to jointly handle anomaly detection and robust time series forecasting within a single, unified framework is highly novel and valuable, addressing a critical limitation of existing works that often assume clean data.

2. The proposed methodology, which strategically employs the Information Bottleneck for detection and Optimal Transport for imputation, is theoretically well-grounded and logically sound, providing a strong, principled foundation for the model's robustness.

3. The experimental section is relatively comprehensive, covering various real-world datasets and successfully demonstrating the superiority of the proposed RockTS model over several strong baseline models, particularly under conditions of data contamination.

**Weaknesses:**

1. The ablation study is thorough but could be strengthened by replacing the Information Bottleneck and Optimal Transport components with more recent and advanced alternatives from the literature on time series anomaly detection and imputation, respectively, to better validate the specific design choices of RockTS.

2. The necessity of the imputation step should be more rigorously justified; specifically, the authors should explore whether irregular time series forecasting models could be directly applied to the sequences after masking anomalies, given that Optimal Transport does not introduce new information. Furthermore, the authors should discuss the potential impact of the OT-based imputation method on the general field of irregular time series prediction.

3. Given that the paper is submitted to ICLR 2026, the comparison against baselines like iTransformer (a 2024 ICLR submission) appears slightly dated; the authors must include comparisons with more recent and state-of-the-art models published in major conferences in 2024 and 2025 to ensure a fair and contemporary evaluation.

**Questions:**

Please refer to Weaknesses.

---

> ### Author Response · Authors · 2025-11-21
>
> We would like to sincerely thank Reviewer **iSu2** for insightful comments and constructive feedback. We have addressed each point below and will revise our paper accordingly.
>
> **Q1:** Strengthen ablation study: replace Information Bottlenecks and Optimal Transportation components with more  advanced alternatives of  time series anomaly detection and imputation, respectively.
>
> **A1:** **Our framework innovatively integrates detection, reconstruction, and prediction through a unified optimization objective, ensuring all components collectively enhance prediction accuracy.**
>
> We add experiments replacing IB-Detector and OT-Reconstruction with the recently developed temporal foundation model Timer [1] for anomaly detection (essentially reconstruction-based anomaly detection) and interpolation in zero-shot setting, respectively.  Moreover, we add the standard anomaly detection dataset WADI and stock dataset with more irregularities NASDAQ Apple Inc (AAPL) to further evaluate the robustness of RockTS.
>
> As shown in the table below, RockTS demonstrates a significant advantage in prediction performance compared to a pipeline performing advanced anomaly detection, interpolation, and prediction independently.
>
> |  | ETTh1 | ETTm2 | WADI | AAPL |
> | --- | --- | --- | --- | --- |
> |  | MSE/MAE | MSE/MAE | MSE/MAE | MSE/MAE |
> | RockTS | **0.399 / 0.417** | **0.248 / 0.303** | **0.520 / 0.137** | **0.018 / 0.033** |
> | advanced anomaly detection + imputation | 0.411 / 0.425 | 0.255 / 0.311 | 0.533 / 0.147 | 0.019 / 0.035 |
>
> **Q2:** Whether irregular time series forecasting models could be directly applied to the sequences after masking anomalies. Furthermore, the authors should discuss the potential impact of the OT-based imputation method on the general field of irregular time series prediction.
>
> **A2:**
>
> **Applicability of Irregular Forecasting Models to Masked Sequences**
>
> The irregular time series forecasting model assumes missing values are random or independent. Directly applying a irregular model would treat masked regions as simple data gaps, failing to leverage detected anomaly information. Our OT reconstruction module considers the anomaly probability determined by the detector during training, which facilitates reconstructing normal complex patterns.
>
> **Impact of OT-Based Imputation on Irregular Time Series Prediction**
>
> Existing irregular time series forecasting methods generally treat all missing points identically, while OT-reconstruction considers prior information about the missing regions within cost function. In this paper, the prior information derives from the anomaly probability assessed by the IB detector. This approach can be extended to irregular time series forecasting by designing cost matrices based on point importance, sensor reliability metrics, or other domain-specific rules.
>
> **Q3:** Comparisons with more recent baselines.
>
> **A3:** We add  the  **up-to-dated methods** CATS(2024), DeformableTST (2024)  and Timemixer++ (2025)[2] as baselines to further evaluate the performance of RockTS. As show in Table below, RockTS shows outstanding performance, especially in WADI and AAPL dataset which have more irregularities or anomalies.
>
> |  | RockTS | CATS | DeformableTST | Timemixer++ |
> | --- | --- | --- | --- | --- |
> |  | MSE/MAE | MSE/MAE | MSE/MAE | MSE/MAE |
> | ETTh1 | **0.399 / 0.417** | 0.411 / 0.431 | 0.417 / 0.434 | 0.437 / 0.449 |
> | ETTm2 | 0.248 / **0.303** | **0.246** / 0.308 | 0.264 / 0.320 | 0.261 / 0.321 |
> | WADI | **0.520 / 0.137** | 0.529 / 0.148 | 0.543 / 0.154 | 0.530 0.151 |
> | AAPL | **0.018 / 0.033** | 0.023 / 0.040 | 0.023 / 0.043 | 0.024 / 0.045 |
>
> [1] Liu, Y., Zhang, H., Li, C., Huang, X., Wang, J., and Long, M. Timer: Transformers for time series analysis at scale. In ICML, 2024.
>
> [2] Wang, S., Li, J., Shi, X., Ye, Z., Mo, B., Lin, W., ... & Jin, M. Timemixer++: A general time series pattern machine for universal predictive analysis. In ICLR, 2025.

---

### Official Review · Reviewer_AifR · 2025-10-30

**Soundness:** 2
**Presentation:** 3
**Contribution:** 1
**Rating:** 4
**Confidence:** 4

**Summary:**

This paper proposes RockTS, an end-to-end framework for robust time series forecasting that addresses the challenge of anomalous subsequences in real-world data. RockTS integrates an Information Bottleneck-based detector to identify and mask anomalous subsequences, and an Optimal Transport-based reconstruction module to impute these regions with normal patterns. The entire process is jointly optimized with the forecasting task, resulting in improved robustness and prediction accuracy. Experiments on real and synthetic datasets demonstrate the superior performance of RockTS over existing methods.

**Strengths:**

S1. RockTS is the first end-to-end framework that directly tackles anomalous subsequences in time series forecasting.
S2. It introduces an adaptive Information Bottleneck-based detector to identify and retain only forecasting-relevant regions.
S3. The Optimal Transport-based reconstruction strategy effectively imputes masked regions while preventing the re-emergence of anomalies.

**Weaknesses:**

W1. The claimed novelty is limited: RockTS mainly composes an IB-based mask and an OT-based imputer on top of existing predictors, so theoretical or algorithmic breakthroughs over prior denoising/imputation work are unclear.

W2. The mask and reconstruction are tightly coupled, so mask errors directly propagate to imputation and forecasting, yet the paper lacks analysis of robustness to wrong masks.

W3. The paper favors hard binary masks via Gumbel-Softmax but provides no direct empirical comparison to soft/probabilistic weighting, leaving the necessity of discretization unproven.

W4. The necessity of OT-based reconstruction is not fully justified. The authors do not compare to simply deleting masked subsequences or using much simpler imputation strategies across tasks.

W5. The training objective leverages future targets to shape masks (IB predictiveness term), raising concerns about train–inference mismatch and potential information leakage that may inflate reported gains.

**Questions:**

Q1. Can the authors clarify and quantify RockTS’s theoretical or empirical advantages over prior denoising+imputation combinations, and highlight any unique insights beyond composing IB and OT?

Q2. How sensitive is end-to-end performance to mask false positives/negatives; can you provide a robustness study or mitigation (e.g., uncertainty-aware masks, mask calibration)?

Q3. Can you report experiments replacing hard binary masks with soft probability weights (and hybrids) to show whether discretization indeed improves forecasting and imputation?

Q4. Have you compared OT reconstruction to the simple alternative of dropping masked segments or to lightweight imputers, and what is the marginal benefit versus computational cost?

Q5. How do you prevent train-time use of future y from creating leakage or unrealistic masks; what are results if masks are learned without access to y or with strictly causal training?

---

> ### Author Response · Authors · 2025-11-21
>
> We would like to sincerely thank Reviewer **AifR** for insightful comments and constructive feedback. We have addressed each point below and will revise our paper accordingly.
>
> **Q1:** Clarify and quantify RockTS’s theoretical or empirical advantages over prior denoising+imputation combinations, and highlight any unique insights beyond composing IB and OT.
>
> **A1:** **Our framework innovatively integrates detection, reconstruction, and prediction through a unified optimization objective, ensuring all components collectively enhance prediction accuracy**. The IB-Detector, based on the information bottleneck principle, unsupervised learns and preserves information useful for prediction. The OT-Reconstruction employs a cost-aware mechanism to effectively prevent the reintroduction of anomalous patterns.
>
> In Table 4, RockTS achieves a significant reduction in mean squared error (MSE) on average compared to traditional independent detection, imputation and prediction processes.
>
> **Our most core unique insight lies in the collaborative mechanism between components.**  Unlike a simple combination of IB and OT, RockTS's detector determines masking based on predicted utility, while the reconstruction module performs imputation according to temporal patterns and detection confidence, while the predictor's gradient simultaneously optimizes both. This collaborative system fundamentally differs from traditional pipeline methods which optimize modules independently and cannot fully serve prediction tasks.
>
> **Q2:** Analysis of robustness to wrong masks. How sensitive is end-to-end performance to mask false positives/negatives; provide a robustness study or mitigation.
>
> **A2:** **RockTS's identification of anomalous subsequences is prediction-oriented**; we use the IB-based detector to adaptively detect and mask regions that are useless for prediction and retain the parts that are relevant for prediction. **Therefore, the regions we identify have different meanings than the anomalous labels.**
>
> To evaluate the model's robustness to the masked regions, we modify the masked regions to respectively test the model's performance when randomly adding 5% masking (setting $\lambda_i$ to 0) and randomly removing 5% masked regions (setting $\lambda _ i$ to 1).  Moreover, we add the standard anomaly detection dataset WADI and stock dataset with more irregularities NASDAQ Apple Inc (AAPL) to further evaluate the robustness of RockTS
>
> The results are shown in the table below. The model results only experienced a slight decline, indicating that the model is robust to minor changes of masked regions.
>
> ||ETTh1|ETTm2|WADI|AAPL|
> |---|---|---|---|---|
> ||MSE/MAE|MSE/MAE|MSE/MAE|MSE/MAE|
> |RockTS|**0.399/0.417**|**0.248/0.303**|**0.520/0.137**|**0.018/0.033**|
> |Randomly remove the mask|0.400/0.417|0.251/0.306|0.519/0.135|0.019/0.035|
> |Randomly add the mask|0.405/0.424|0.254/0.307|0.521/0.137|0.019/0.035|
>
> **Q3:** Replacing hard binary masks with soft probability weights (and hybrid models) to validate whether discretization genuinely indeed improves forecasting and imputation.
>
> **A3: Soft masks fail to completely isolate anomalous regions, allowing residual contamination to propagate to the forecasting module,** while hard masks provide clean separation between anomalous and normal regions, enabling more effective reconstruction.
>
> We supplement the experiment by replacing the masks with a continuous weight vector $\lambda \in [0,1]^L$ and setting $\mathbf{x}_\text{m} = \mathbf{x} \odot \lambda$. Experimental results demonstrate that hard binary masks significantly reduce prediction errors compared to soft probabilistic weighting.
>
> ||ETTh1|ETTm2|WADI|AAPL|
> |---|---|---|---|---|
> ||MSE/MAE|MSE/MAE|MSE/MAE|MSE/MAE|
> |hard binary masks (ours)|**0.399/0.417**|**0.248/0.303**|**0.520/0.137**|**0.018/0.033**|
> |soft probability weights|0.400/0.420|0.251/0.306|0.525/0.144|0.020/0.038|

---

> ### Author Response · Authors · 2025-11-21
>
> **Q4:** Compare OT reconstruction to the simple alternative of dropping masked segments or to lightweight imputers, and what is the marginal benefit versus computational cost?
>
> **A4: Dropping masked segments disrupts the continuity and periodicity of time series, and simple approaches such as linear interpolation cannot reconstruct complex nonlinear normal patterns.** Our OT reconstruction module considers the attention matrix among time points and the anomaly probability determined by the detector during training, which facilitates reconstructing normal complex patterns.
>
> We add experiments replacing OT reconstruction with dropping masked segments (padded with zeros before the series to maintain its total length) and linear interpolation. As shown in the table below, demonstrating that OT reconstruction exhibits significant advantages:
>
> ||ETTh1|ETTm2|WADI|AAPL|
> |---|---|---|---|---|
> ||MSE/MAE|MSE/MAE|MSE/MAE|MSE/MAE|
> |OT-Reconstruction (ours)|**0.399/0.417**|**0.248/0.303**|**0.520/0.137**|**0.018/0.033**|
> |Dropping masked segments|0.444/0.456|0.269/0.329|0.543/0.153|0.022/0.042|
> |Linear interpolation|0.408/0.421|0.253/0.306|0.542/0.152|0.022/0.044|
>
> **Computational Cost Analysis:**
> OT reconstruction increases the overall inference time by 0.14-0.25 (using ETTh1 as an example) compared to dropping masked segments and linear interpolation. However, the absolute cost remains practical:
>
> ||RockTS|Dropping masked segments|Linear interpolation|
> |---|---|---|---|
> |inference time (sec/iter)|0.0074|0.0059|0.0065|
>
> **Q5:**  Concerns about train–inference mismatch and potential information leakage.
>
> **A5:** **We would like to clarify that our framework does not suffer from information leakage or a train-inference mismatch.**
>
> The future target y is solely used to compute the prediction loss L_pred, which optimizes model parameters to guide the detector in learning which patterns possess persistent predictive power and should be retained. It does not learn to generate customized masks tailored to the specific y value of the current sample. **At no stage is $\mathbf{y}$ ever used as input for the mask generation process.**
>
> **Prediction loss is indispensable to the entire framework** because our model is not trained to exclude specific anomalous patterns. Instead, it automatically identifies and masks elements of uncertainty that disrupt predictions through the principle of information bottlenecks, thereby learning robust representations of normal signals.

---

### Official Review · Reviewer_4eBJ · 2025-11-01

**Soundness:** 3
**Presentation:** 3
**Contribution:** 2
**Rating:** 6
**Confidence:** 3

**Summary:**

This paper proposes RockTS, a robust method for time series forecasting. Specifically, the authors adopt a detection-imputation pipeline. First, anomalies in the time series are detected based on the Information Bottleneck principle; then, the anomalous segments are imputed using normal patterns derived from optimal transport. Experiments on both synthetic and real-world datasets demonstrate that RockTS achieves better performance than baseline methods.

**Strengths:**

1. The proposed methods are both sound and novel, as they employ the Information Bottleneck principle to detect anomalies in time series.

2. Visualization results demonstrate that RockTS has strong reconstruction capabilities for anomalous time series.

3. Experiments indicate consistent and significant improvements over all baseline methods.

**Weaknesses:**

1. In the experiments, the authors primarily compare methods designed for general time series forecasting, rather than those specifically developed for robust time series forecasting, such as the robust forecasting approaches discussed in the related work section.

2. There is a lack of ablation studies on hyper-parameters, such as the value of $\alpha$ in Equation (15).

**Questions:**

1. For the imputation part, what would be the outcome if we impute the time series using simple approaches such as linear interpolation? I raise this question because it can help investigate whether detection or imputation is more important for robust time series forecasting.

---

> ### Author Response · Authors · 2025-11-21
>
> We would like to sincerely thank Reviewer **4eBJ** for insightful comments and constructive feedback. We have addressed each point below and will revise our paper accordingly.
>
> **Q1:** Comparisons with more robust time series forecasting approaches.
>
> **A1:** We add  the relevant method RobustTSF [1] (mainly considers point anomalies)  and TimeBridge [2] (considers non-stationarity) to further evaluate the performance of RockTS.  Moreover, we add the standard anomaly detection dataset WADI and stock dataset with more irregularities NASDAQ Apple Inc (AAPL) to further evaluate the robustness of RockTS. As show in Table below, RockTS shows outstanding performance, especially in WADI and AAPL dataset which have more irregularities or anomalies.
>
> |  | RockTS | RobustTSF | TimeBridge |
> | --- | --- | --- | --- |
> |  | MSE/MAE | MSE/MAE | MSE/MAE |
> | ETTh1 | **0.399 / 0.417** | 0.428 / 0.447 | 0.422 / 0.440 |
> | ETTm2 | **0.248 / 0.303** | 0.263 / 0.323 | 0.253 / 0.307 |
> | WADI | **0.520 / 0.137** | 0.550 / 0.152 | 0.580 / 0.149 |
> | AAPL | **0.018 / 0.033** | 0.023 / 0.040 | 0.021 / 0.041 |
>
> **Q2:** There is a lack of ablation studies on hyper-parameters, such as the value of $\alpha$  in Equation (15).
>
> **A2:** In Appendix D, we present experimental results for different values of the hyperparameters $\alpha$ and $\beta$ used to balance the loss function, as well as the masking sparsity parameter $\tau$.
>
> **Q3:** For the imputation part, what would be the outcome if we impute the time series using simple approaches such as linear interpolation?
>
> **A3:**
>
> **Simple approaches such as linear interpolation cannot reconstruct complex nonlinear normal patterns.** Our OT reconstruction module considers the attention matrix among time points and the anomaly probability determined by the detector during training, which facilitates reconstructing normal complex patterns.
>
> We add experiments replacing OT reconstruction with linear interpolation. As shown in the table below, using linear interpolation for imputation leads to a significant performance decline, clearly demonstrating that imputation and detection are equally important for improving prediction accuracy.
>
> |  | ETTh1 | ETTm2 | WADI | AAPL |
> | --- | --- | --- | --- | --- |
> |  | MSE/MAE | MSE/MAE | MSE/MAE | MSE/MAE |
> | OT-Reconstruction | **0.399 / 0.417** | **0.248 / 0.303** | **0.520 / 0.137** | **0.018 / 0.033** |
> | Linear interpolation | 0.408 / 0.421 | 0.253 / 0.306 | 0.542 / 0.152 | 0.022 / 0.044 |
>
> [1] Cheng, H., Wen, Q., Liu, Y., & Sun, L. (2024). Robusttsf: Towards theory and design of robust time series forecasting with anomalies. *arXiv preprint arXiv:2402.02032*.
>
> [2] Liu, P., Wu, B., Hu, Y., Li, N., Dai, T., Bao, J., & Xia, S. T. (2024). Timebridge: Non-stationarity matters for long-term time series forecasting. *arXiv preprint arXiv:2410.04442*.

---

### Official Review · Reviewer_yGLS · 2025-11-01

**Soundness:** 2
**Presentation:** 3
**Contribution:** 2
**Rating:** 4
**Confidence:** 3

**Summary:**

This work proposes an end-to-end robust forecasting framework RockTS that integrates anomalous-subsequence detection and imputation directly into the forecasting objective. It employs an Information-Bottleneck (IB) mask to eliminate prediction-irrelevant segments and uses an Optimal-Transport (OT) refill to reconstruct the masked areas into normal patterns while preventing the reappearance of anomalies.

**Strengths:**

1. It is reasonable to explicitly incorporate anomalous subsequences into the main objective of time series prediction, rather than removing them during preprocessing or relying on robust loss functions.

2. The combination of Information Bottleneck (IB) to generate masks based on predictive correlations and Optimal Transport (OT) to fill in values and prevent abnormal reproduction is novel.

3. The paper is well organized and presents the method with clarity.

**Weaknesses:**

1. Despite being proposed as a robust method, it has not been evaluated under a wide range of abnormal scenarios, such as varying noise levels and different lengths of anomalous segments.

2. The baseline comparison primarily includes standard forecasting models, while robustness-oriented methods are represented only by TAFAS.

3. The paper reports inference time per sample but does not provide a theoretical analysis of the algorithm’s time complexity.

**Questions:**

1. Additional experiments with robust baselines under diverse noise conditions are needed to validate the claimed robustness.

2. Which specific components or stages are included in the reported inference time per sample?

3. The paper should include a theoretical analysis of the model’s time complexity.

4. Given that the appendix claims the model can be deployed in a real-time environment, what is its memory overhead on GPU and CPU?

---

> ### Author Response · Authors · 2025-11-21
>
> We would like to sincerely thank Reviewer **yGLS** for insightful comments and constructive feedback. We have addressed each point below and will revise our paper accordingly.
>
> **Q1:** It has not been evaluated under a wide range of abnormal scenarios, such as varying noise levels and different lengths of anomalous segments.
>
> **A1:** To further validate RockTS's robustness under various anomaly configurations, we add experiments with different anomaly types and strength.
>
> **Distinguish the types of anomalies：**
>
> We inject six different exceptions at 10% each on the ETTh1 dataset to test which exceptions our proposed method and baseline are good at handling respectively.
>
> With an input length of 512 and an output length of 96, the following table shows the performance of the different methods on different types of anomalies. RockTS exhibits significant advantages over the baseline under all six anomaly categories, which demonstrates that our proposed IB-based Detector can be applied to diverse anomalies.
>
> ||RockTS|ITransformer|PatchTST|Pathformer|TimesNet|ModernTCN|Dlinear|TiDE|FITS|TimeMixer|
> |---|---|---|---|---|---|---|---|---|---|---|
> ||MSE/MAE|MSE/MAE|MSE/MAE|MSE/MAE|MSE/MAE|MSE/MAE|MSE/MAE|MSE/MAE|MSE/MAE|MSE/MAE|
> |oulter|0.390/0.395|0.424/0.433|0.402/0.416|0.469/0.455|0.412/0.420|0.404/0.413|0.407/0.417|0.426/0.432|0.397/0.409|0.402/0.417|
> |hmirror|0.481/0.440|0.519/0.475|0.493/0.460|0.649/0.536|0.496/0.463|0.496/0.457|0.498/0.464|0.510/0.466|0.493/0.455|0.497/0.457|
> |vmirror|0.442/0.429|0.472/0.458|0.450/0.445|0.578/0.509|0.470/0.450|0.451/0.441|0.452/0.444|0.468/0.455|0.443/0.437|0.460/0.452|
> |scale|0.452/0.426|0.506/0.475|0.463/0.439|0.661/0.533|0.532/0.482|0.471/0.446|0.467/0.447|0.493/0.463|0.465/0.444|0.478/0.456|
> |pattern|0.501/0.474|0.687/0.554|0.665/0.532|0.736/0.563|0.771/0.577|0.673/0.539|0.658/0.543|0.691/0.553|0.666/0.537|0.674/0.539|
> |noise|0.367/0.392|0.409/0.429|0.388/0.412|0.446/0.447|0.401/0.420|0.388/0.410|0.391/0.415|0.407/0.427|0.382/0.406|0.387/0.412|
> |original| 0.361/0.386|0.386/0.405| 0.377/0.397|0.372/0.392|0.470/0.470|0.377/0.402|0.375/0.399|0.393/0.418|0.376/0.396|0.372/0.401|
>
> **Distinguishing the strength of  of anomalies：**
>
> We test prediction performance between RockTS and PatchTST by injecting pattern anomalies with lengths ranging from 5 to 50. As shown in the following table, RockTS presents a slight advantage over PatchTST when the length of pattern is 5, and then the advantage of RockTS gradually increases as the strength of anomalies increases.
> In the revised version of the article, we will complement the analysis of each anomaly.
>
> |Pattern_length|5|10|20|30|40|50|
> |---|---|---|---|---|---|---|
> |Metric|MSE/MAE|MSE/MAE|MSE/MAE|MSE/MAE|MSE/MAE|MSE/MAE|
> |PatchTST|0.403/0.419|0.512/0.478|0.578/0.514|0.601/0.526|0.645/0.539|0.635/0.533|
> |RockTS|0.401/0.414|0.458/0.455|0.521/0.486|0.566/0.499|0.561/0.497|0.587/0.513|
>
> **Q2:** Lack of baseline comparison with robustness-oriented methods.
>
> **A2:** We add the relevant method RobustTSF (mainly considers point anomalies)  and TimeBridge (considers non-stationarity) to further evaluate the performance of RockTS.  Moreover, we add the standard anomaly detection dataset WADI and stock dataset with more irregularities NASDAQ Apple Inc (AAPL) to further evaluate the robustness of RockTS. As show in Table below, RockTS shows outstanding performance, especially in WADI and AAPL dataset which have more irregularities or anomalies.
>
> ||RockTS|RobustTSF|TimeBridge|
> |---|---|---|---|
> ||MSE/MAE|MSE/MAE|MSE/MAE|
> |ETTh1|0.399/0.417|0.428/0.447|0.422/0.440|
> |ETTm2|0.248/0.303|0.263/0.323|0.253/0.307|
> |WADI|0.520/0.137|0.550/0.152|0.580/0.149|
> |AAPL|0.018/0.033|0.023/0.040|0.021/0.041|

---

> ### Author Response · Authors · 2025-11-21
>
> **Q3:** Theoretical analysis of the algorithm’s time complexity.
>
> **A3:** To better analyze the time complexity, we provide the algorithm's pseudocode as follows:
>
> **Algorithm: RockTS Framework:**
>
> Input: Historical time series  $\mathbf{x} =( {x_1, \dots, x_L})$,
>
> Look-back Window: $L$, Forecast horizon  $F$,
>
> Hyperparameters:  $\alpha, \beta, \tau$,
>
> Output: Forecasted series   $\mathbf{\hat{y}} = (\{\hat{x}\_{L+1}, \dots, \hat{x}\_{L+F}\})$.
>
> **Phase 1: Anomaly Detection via Information Bottleneck**
>
> 1: Encode  $\mathbf{x}$ to latent features: $\mathbf{Z} = \text{Linear}(\mathbf{x}) \in \mathbb{R}^{L \times D}$
>
> 2: Compute self-attention:
>
> $\mathbf{Q} = \mathbf{W_Q Z}, \mathbf{K} = \mathbf{W_K Z}, \mathbf{V} = \mathbf{W_V Z}$
>
> $\mathbf{A} = \text{Softmax}\left(\frac{\mathbf{QK}^\top}{\sqrt{D}}\right)$
>
> $\mathbf{E} = \mathbf{AV}$
>
> 3: Generate mask probabilities via cross-attention:
>
> $\lambda = \sigma\left(\text{CrossAttention}(\mathbf{E}, \mathbf{Z}) \cdot \mathbf{W_b}^\top\right),  \lambda \in [0, 1]^L$
>
> 4: Sample binary mask with Gumbel-Softmax:
>
> $\mathbf{M} = \text{GumbelSoftmax}(\lambda)  , m_i \in \{0, 1\}$
>
> 5: Apply mask: $\mathbf{x_m} = \mathbf{x} \odot \mathbf{M}$
>
> 6.Compute mask sparsity loss :
>
> $\mathcal{L} _ \text{M} = \sum_{i=1}^L \left[ \lambda_i \log\frac{\lambda_i}{\tau} + (1-\lambda_i)\log\frac{1-\lambda_i}{1-\tau} \right] + \frac{1}{L} \sum_{i=1}^{L-1} |\lambda_{i+1} - \lambda_i|$
>
> **Phase 2: Optimal Transport-based Reconstruction**
>
> 7: Generate initial reconstruction:
>
> $\widetilde{\mathbf{x}} = \mathcal{G}(\mathbf{x_m}), \mathcal{G}$: Transformer encoder + linear head
>
> 8: Compute transport matrix: $\mathbf{P} = \text{MLP}(\mathbf{A})$
>
> 9: Reconstruct with OT constraints: $\mathbf{x'} = \text{softmax}(\mathbf{P})^\top \widetilde{\mathbf{x}}$
>
> 10: Calculate transport loss and the optimal transport loss:
>
> $\mathbf{C}\_{i, j} = \left \lbrace \begin{array}{cc}
> 1-\lambda_j, & \mathbf{M}_j=0 \\\\
> 0, & \mathbf{M}\_j=1
> \end{array}\right.$
>
> $\mathcal{L}\_\text{OT} = \frac{1}{L} \|\mathbf{x} - \mathbf{x'}\|^1 + \beta \sum\_{i,j} \mathbf{P}\_{i,j} \odot \mathbf{C}\_{i,j}$
>
> **Phase 3: Forecasting**
> 11: Segment $\mathbf{x'}$into patches, and encode patches with position embeddings:
>
> $\mathbf{Z'} = \mathbf{W\_p}\text{Patchify}(\mathbf{x'} )+ \mathbf{W\_{pos}}$
>
> 12: Generate predictions via Transformer:
>
> $\mathbf{\hat{y}} = \text{Transformer}(\mathbf{Z'})$
>
> 13: Compute prediction loss:
>
> $\mathcal{L}_{\text{pred}} = \|\mathbf{y} - \mathbf{\hat{y}}\|_F^2$
>
> **Optimization**
>
> 14: Total loss:
>
> $\mathcal{L} = \alpha \mathcal{L}-\text{M} + \mathcal{L}\_\text{OT} + \mathcal{L}\_{\text{pred}}$
>
> 15: Update parameters via backpropagation
>
> Since Phase 1 requires point-level attention calculations, the overall time complexity of the algorithm will reach $O(L^2)$.
>
> We acknowledge that the $O(L^2)$ complexity is a theoretical limitation of our current method, particularly when dealing with extremely long sequences. However, for mainstream time-series forecasting tasks, input lengths are typically limited to a range of 96 to 720 points. Within this scope, the computational cost of  $O(L^2)$ is acceptable during both training and inference phases, as our experiments in Appendix 8 have demonstrated.
>
> **Q4:** Which specific components or stages are included in the reported inference time per sample？
>
> **A4:** It includes the following components: 1. Identifying the anomalous sub-sequences using the IB-based Detector; 2. Imputing these detected regions via the OT reconstruction module; 3. Generating the final prediction results through the prediction module.
>
> **Q5:** The memory overhead of this algorithm on GPUs and CPUs.
>
> **A5:** With batch size of 1, input length of 512, predicted length of 96, and 1 channel, a single inference operation consumes approximately 500MB of GPU memory and approximately 2GB of CPU memory.

---

### Official Review · Reviewer_HGzc · 2025-11-03

**Soundness:** 2
**Presentation:** 3
**Contribution:** 2
**Rating:** 4
**Confidence:** 4

**Summary:**

Real-world time series data often contain anomalous subsequences that deviate from the regular patterns of the entire sequence, posing challenges for accurate forecasting. This paper proposes a novel end-to-end framework for robust time series forecasting called RockTS, which first introduces an anomaly pattern detection process based on the information bottleneck, which compresses the representation of the time series while retaining information more relevant to effective forecasting. Then, it reconstructs the detected anomalous regions into normal patterns  based on optimal transport for prediction.

**Strengths:**

(1) It introduces an adaptive detector based on the information bottleneck to detect anomalous subsequences in the prediction process and retain prediction-relevant regions.

(2) It designs a reconstruction strategy based on optimal transport (OT) to fill in the detected anomalous regions with normal data patterns.

(3) Experiments show that ROCKTS outperforms classic time series forecasting algorithms, and ablation experiments demonstrate the effectiveness of the proposed modules.

**Weaknesses:**

(1) The code implementation does not seem to align well with the paper's description. For example, the paper mentions using MSE loss, but it appears that MAE loss is used in the code.

(2) There are concerns about unfair comparisons. (i) It seems that some methods' results are referenced from published papers, but most of them use MSE loss, while the authors' method uses MAE loss. (ii) Some results differ significantly from the published methods at length 512. For example, the average of 512-PatchTST-ETTh1 over four prediction lengths is 0.331 form its paper, while the authors report 0.351.

(3) The baselines used are relatively old, and there is a lack of more recent baselines, such as CATS and DeformableTST.

(4) The advantages on large-scale datasets such as Electricity, traffic, and weather are not obvious. For example, on traffic, it is even worse than PatchTST. Since the authors' code is mainly modified from PatchTST, this suggests that the newly added modules may be harmful for some datasets. After artificially injecting noise into the dataset, the accuracy of ROCKTS deteriorates severely compared to the original ROCKTS. If the ROCKTS modules were truly effective, such a significant deterioration should not occur. Based on the above analysis, considering that the Gumbel softmax is unstable and difficult to optimize, and there are no explicit supervision information for anomaly detection in the paper, this raises concerns about the effectiveness and generalizability of the IB-based Detector.

(5) Anomaly detection is entirely unsupervised. We do not know whether the current sample contains anomalies. It is highly likely that normal patterns are mistakenly corrected. This may also be why it performs worse than the baseline model on large-scale datasets. Moreover, the current method forces the model to identify anomalies, which may not be reasonable and could lead to the risk of performance degradation.

(6) In terms of efficiency, compared with patchTST, the inference overhead nearly doubled on the weather, electricity, and traffic datasets, while the accuracy remained the same or even poorer than PatchTST. The efficiency trade-off does not seem to have yielded the expected performance improvement.

**Questions:**

The output mask of Gumbel softmax lacks explicit supervision information for anomalies and non-anomalies. Even under the implicit constraints of mutual information theory, it seems hard to imagine that it can effectively detect truly anomalous subsequences? For Gumbel softmax, it seems highly likely that normal patterns are misjudged as anomalies, leading to a deterioration in model performance.

---

> ### Author Response · Authors · 2025-11-21
>
> We would like to sincerely thank Reviewer **HGzc** for insightful comments and constructive feedback. We have addressed each point below and will revise our paper accordingly.
>
> **Q1:** The paper mentions using MSE loss, but it appears that MAE loss is used in the code.
>
> **A1:** Thank you for pointing out the error in our wording. In our implementation, we found MAE to be more stable during training, especially when dealing with anomalous subsequences. We will correct the paper to reflect the use of MAE loss and provide a detailed explanation in the revised version.
>
> **Q2-1:** Most of the baseline use MSE loss, while the authors' method uses MAE loss.
>
> **A2-1:** In comparative experiments against other baseline models, we retained their default loss function selections.
>
> To address your concerns regarding unfair comparisons, we select the three top-performing models (PatchTST, Pathformer, and FITS) and supplement the results with training using the MAE loss function. Moreover, we add the standard anomaly detection dataset WADI and stock dataset with more irregularities NASDAQ Apple Inc (AAPL) to further evaluate the robustness of RockTS. As shown in the table below, RockTS still holds an advantage over the baselines.
>
> ||RockTS|PatchTST|Pathformer|FITS|
> |---|---|---|---|---|
> ||MSE/MAE|MSE/MAE|MSE/MAE|MSE/MAE|
> |ETTh1|**0.399/0.417**|0.413/0.429|0.416/0.426|0.405/0.424|
> |ETTm2|**0.248/0.303**|0.255/0.310|0.253/0.305|0.252/0.310|
> |WADI|**0.520/0.137**|0.555/0.161|0.557/0.164|0.565/0.166|
> |AAPL|**0.018/0.033**|0.024/0.047|0.023/0.045|0.025/0.047|
>
> **Q2-2:** Some results differ significantly from the published methods at length 512.
>
> **A2-2:** For the purpose of fair comparison, we addressed the “drop-last” issue [1] present in the code framework of the methods (such as patchtst, as you mentioned) and reran them. Consequently, the reported results differ from the original paper.
>
> **Q3:** Comparisons with more recent baselines.
>
> **A3:** We add the up-to-dated methods CATS(2024), DeformableTST (2024) and Timemixer++ (2025) [2] as baselines to further evaluate the performance of RockTS. As show in Table below, RockTS shows outstanding performance, especially in WADI and AAPL dataset which have more irregularities or anomalies.
>
> ||RockTS|CATS|DeformableTST|Timemixer++|
> |---|---|---|---|---|
> ||MSE/MAE|MSE/MAE|MSE/MAE|MSE/MAE|
> |ETTh1|**0.399/0.417**|0.411/0.431|0.417/0.434|0.437/0.449|
> |ETTm2|0.248/**0.303**|**0.246**/0.308|0.264/0.320|0.261/0.321|
> |WADI|**0.520/0.137**|0.529/0.148|0.543/0.154|0.530/0.151|
> |AAPL|**0.018/0.033**|0.023/0.040|0.023/0.043|0.024/0.045|
>
> **Q4-1**: The advantages on large-scale datasets such as Electricity, traffic, and weather are not obvious.
>
> **A4-1:** Real-world time series datasets often contain varying degrees of anomalous sub-sequences or irrelevant information. On average, our approach achieves significant improvements across diverse datasets by effectively mitigating the impact of such disturbances. Furthermore, even when anomalies or irrelevant information are entirely absent or minimal, our design ensures that only components with the lowest relevance to the prediction task are filtered out. Therefore, the performance degradation in minority cases remains within manageable limits.
>
> **Q4-2:** After artificially injected noise into the dataset, the accuracy of ROCKTS deteriorates severely compared to the original ROCKTS.
>
> **Q4-2:** The performance drop under heavy noise injection is expected, because: 1. The effective information content of the model inputs has been reduced. 2. The impact of such noise is difficult to completely avoid.
>
> Our approach mitigates the impact of noise and demonstrates significantly better performance than the baseline under noisy conditions, which demonstrates its effectiveness and value.

---

> ### Author Response · Authors · 2025-11-22
>
> **Q5:** Anomaly detection is entirely unsupervised. For Gumbel softmax, it seems highly likely that normal patterns are misjudged as anomalies, leading to a deterioration in model performance.
>
> **A5: RockTS's identification of anomalous subsequences is prediction-oriented.** We use the IB-based detector to adaptively detect and mask regions that are less useful for forecasting, which may include both anomalies and non-informative normal points. **Therefore, the regions we identify have different meanings than the anomalous labels.**
>
> Although the masking is unsupervised, **it is directly tied to the forecasting objective, which aligns with the goal of robust prediction**. Our ablation studies confirm that both the IB-based detector and OT-based reconstruction contribute positively to the overall performance.
>
> **Q6:** The efficiency trade-off does not seem to have yielded the expected performance improvement.
>
> **A6:** We acknowledge that RockTS incurs higher inference time due to the detection and imputation modules. However, for mainstream time series forecasting tasks, input lengths are typically limited to a range of 96 to 720 points. Within this scope, the computational cost is acceptable, as our experiments in Appendix 8 have demonstrated.
>
> [1] Qiu, X., Hu, J., Zhou, L., Wu, X., Du, J., Zhang, B., ... & Yang, B. Tfb: Towards comprehensive and fair benchmarking of time series forecasting methods. In VLDB, 2024.
>
> [2] Wang, S., Li, J., Shi, X., Ye, Z., Mo, B., Lin, W., ... & Jin, M. Timemixer++: A general time series pattern machine for universal predictive analysis. In ICLR, 2025.

---

### Author Response · Authors · 2025-11-29
**Summary of the rebuttal**

We thank the Reviewers for the insightful comments and detailed feedback. We are delighted that reviewers find our paper has the following advantages:

- **Novel and Well-Motivated Framework:** RockTS is the first end-to-end framework that integrates anomaly detection and imputation directly into the forecasting objective via a unified optimization process, addressing the issue of anomalous subsequences in time series forecasting. (**iSu2, yGLS, 4eBJ**)
- **Theoretically Grounded Methodology:** The use of Information Bottleneck for adaptive anomaly detection and Optimal Transport for reconstruction is both theoretically sound. (**iSu2, yGLS, 4eBJ**)
- **Strong Empirical Performance:** Comprehensive experiments on multiple real-world and synthetic datasets demonstrate that RockTS consistently outperforms baselines, especially under conditions with higher levels of anomalies. (**iSu2, 4eBJ, HGzc**)

In response to the reviewers' concerns, we have conducted extensive revisions and provided detailed rebuttals:

- **Fairness in Comparisons and More Baselines (HGzc, 4eBJ, yGLS, iSu2) :** We corrected the loss function description (MAE used in practice), reran baselines under consistent settings, and added recent SOTA models (CATS, DeformableTST, Timemixer++) and robustness-focused methods (RobustTSF, TimeBridge). Results confirm RockTS's superiority.
- **Robustness Under Diverse Anomalies (yGLS, HGzc):** We added experiments with varying anomaly types (outlier, mirroring, scaling, pattern, noise) and strengths, showing RockTS consistently outperforms baselines across all settings.
- **Justification of Components (AIfR, 4eBJ, iSu2):** We validated the necessity of both IB-based detection and OT-based reconstruction through analytical experiment, showing that:
    - Hard binary masks outperform soft masks.
    - OT reconstruction significantly outperforms dropping segments or linear interpolation.
    - RockTS significantly outperforms performing advanced anomaly detection and imputation independently.
    - The framework is robust to minor mask errors.
- **Efficiency and Complexity (yGLS, HGzc):** We provided time complexity analysis  ( $O(L^2)$) and memory overhead (GPU/CPU), confirming practicality for typical sequence lengths.
- **Theoretical and Practical Novelty (AIfR):** We emphasized the synergistic optimization of detection, reconstruction, and forecasting—distinguishing RockTS from simple pipeline combinations—and clarified that no information leakage occurs.

We believe our explanation addresses the reviewers' concerns and hope it will be taken into account in the final decision. We would like to express our sincere appreciation for the reviewers and ACs for their time and consideration.

---

### Meta-Review · Area_Chair_5862 · 2025-12-19

**Summary:**

This paper proposes a novel end-to-end framework, RockTS, for robust time series forecasting, based on information bottleneck and optimal transport. This framework integrates anomaly subsequence detection and imputation into the forecasting task through a unified optimization objective. Experimental results demonstrate that RockTS achieves good performance. However, there are some issues regarding the credibility of the experiments, time complexity, and the novelty of the model, which means the current version requires significant improvements before submission in the future at an appropriate time.

**Reviewer Concerns:**

The reviewers raised several key concerns about the paper:

- Implementation and Fairness: Discrepancies were noted between the code implementation and the description in the paper, particularly regarding the use of MAE versus MSE loss. This raises questions about the fairness of baseline comparisons.

- There are doubts about the effectiveness of the IB-based Gumbel Softmax mask, given the lack of explicit supervision. Reviewers are concerned about the potential misidentification of normal patterns as anomalies, especially under various noise conditions.

- The inference time is doubled compared to other methods without significant accuracy improvement or proper complexity analysis, leading to questions about the method’s efficiency and feasibility for deployment.

- Design and Novelty: The method appears to be a composition of existing techniques without clear theoretical or algorithmic advances, which limits its perceived novelty.

- Training Objective Concerns: There is potential for information leakage in the training objective, which might lead to a train-inference mismatch, casting doubt on the reliability of the reported gains.

The author’s response in the rebuttal addresses many concerns, but I believe further clarification is needed in terms of experiments and efficiency. In terms of experiments, the author added two additional baselines during the response phase but missed strong competitors such as DUET, SparseTF, and TimeKAN for time series models. Similarly, the author’s experimental setup, with a lookback window set to 512, is confusing as it differs from existing popular protocols, which usually set it to 96, 168, or 336, and the author did not provide a specific reason. Furthermore, the issue of experimental fairness has not been thoroughly resolved, as all models need to be run under a unified setting. Lastly, although the author claims that the model’s time complexity is tolerable on the current small-scale datasets, it becomes problematic on large-scale datasets, limiting the model’s effectiveness.

Overall, the reviewer raised so many issues indicating that this version of the paper requires significant revisions, and encourages the author to resubmit at an appropriate time.

**Reviewer Scores:**

The initial score for the paper was 66444, but the paper did not receive any replies. After carefully reading the paper, the reviewer comments, and the author’s rebuttal, I believe the paper needs to be revised to meet the high standards of ICLR.

---

### Decision · Program_Chairs · 2026-01-26

Reject